# The PMA phorbol ester tumor promoter increases canonical Wnt signaling via macropinocytosis

Nydia Tejeda-Munoz[1,2†], Yagmur Azbazdar[1†], Julia Monka[1], Grace Binder[1], Alex Dayrit[1], Raul Ayala[3], Neil O'Brien[3], Edward M De Robertis[1*]

[1]Department of Biological Chemistry, David Geffen School of Medicine, University of California, Los Angeles, Los Angeles, United States; [2]Department of Oncology Science, Health Stephenson Cancer Center, University of Oklahoma Health Science Center, Oklahoma City, United States; [3]Department of Medicine, David Geffen School of Medicine, University of California, Los Angeles, Los Angeles, United States

*For correspondence:
ederobertis@mednet.ucla.edu

†These authors contributed equally to this work

Competing interest: The authors declare that no competing interests exist.

**Abstract** Activation of the Wnt pathway lies at the core of many human cancers. Wnt and macropinocytosis are often active in the same processes, and understanding how Wnt signaling and membrane trafficking cooperate should improve our understanding of embryonic development and cancer. Here, we show that a macropinocytosis activator, the tumor promoter phorbol 12-myristate 13-acetate (PMA), enhances Wnt signaling. Experiments using the *Xenopus* embryo as an in vivo model showed marked cooperation between the PMA phorbol ester and Wnt signaling, which was blocked by inhibitors of macropinocytosis, Rac1 activity, and lysosome acidification. Human colorectal cancer tissue arrays and xenografts in mice showed a correlation of cancer progression with increased macropinocytosis/multivesicular body/lysosome markers and decreased GSK3 levels. The crosstalk between canonical Wnt, focal adhesions, lysosomes, and macropinocytosis suggests possible therapeutic targets for cancer progression in Wnt-driven cancers.

## eLife assessment

Altogether, the strength of this **important** study is that it provides **compelling** evidence in several biological models, including *Xenopus* embryos, that Wnt3a increases macropinocytosis and that PMA increases this cellular response. This novel link between Wnt, focal adhesions, lysosomes, and macropinocytosis will be very interesting for cell and tumor biologists. In future work, it will be important to identify the underlying mechanism, i.e., the molecular node whereby this interaction occurs.

## Introduction

Cancer progression is a process during which mutations in various signaling pathways accumulate in the cell leading to uncontrolled growth (*Kinzler and Vogelstein, 1996*; *Hanahan and Weinberg, 2011*). Cancer progression is also influenced by tumor promoters, substances that do not mutate the DNA by themselves but nevertheless facilitate cancer development. In 1941, Berenblum and Rous found that inflammatory agents such as croton oil or turpentine could induce skin cancer when applied repeatedly to mouse or rabbit skin, but only when an initiator mutagen such as benzopyrene or anthracene had been previously painted on that patch of skin (*Berenblum, 1941*; *Rous and Kidd, 1941*). The active agent in croton oil was later purified and found to be the phorbol ester phorbol 12-myristate 13-acetate (PMA), also known as 12-*O*-tetradecanoylphorbol-13-acetate (TPA).

The molecular mechanism by which PMA promotes cancer is incompletely understood but involves the activation of protein kinase C (PKC) in the plasma membrane, where PMA mimics the endogenous second messenger diacylglycerol (DAG) (*Nishizuka, 1984*). Tumor promoters cause chronic inflammation and significantly contribute to tumorigenesis (*Weinberg and Weinberg, 2007*).

The canonical Wnt/β-catenin pathway is a major driver of cancer and plays a crucial role in embryonic development (*MacDonald et al., 2009*; *Nusse and Clevers, 2017*). In the case of colon cancer, the great majority of tumors are initiated by mutation of the tumor suppressor Adenomatous Polyposis Coli (APC), which is a component of the destruction complex that normally degrades the transcriptional regulator β-catenin, leading to its accumulation in the nucleus and benign polyp formation (*Kinzler and Vogelstein, 1996*; *Segditsas and Tomlinson, 2006*). Activation of the Wnt/β-catenin pathway is also the driving force in many other cancers (*Galluzzi et al., 2019*). A key regulator of Wnt signaling is Glycogen Synthetase 3 (GSK3), which phosphorylates β-catenin, leading to the formation of a phosphodegron that triggers its degradation in the proteasome. Upon activation of the Wnt receptors, GSK3, together with the destruction complex, is sequestered into multivesicular bodies (MVBs), also known as the late endosomes (*Taelman et al., 2010*; *Vinyoles et al., 2014*). The translocation of GSK3 from the cytosol into MVBs/lysosomes leads to the stabilization of many other cellular proteins in addition to β-catenin, in a process designated Wnt-STOP (Wnt-Stabilization of Proteins) (*Acebron et al., 2014*; *Albrecht et al., 2021*).

The many GSK3 phosphorylation targets stabilized by Wnt signaling include regulators of macropinocytosis such as Ras, Rac1, and Pak1 (*Taelman et al., 2010*; *Albrecht et al., 2020*; *Jeong et al., 2012*). A recent advance was the realization that Wnt signaling requires macropinocytosis (i.e., cell drinking) in order to traffic activated Wnt receptors and GSK3 to MVBs and lysosomes (*Redelman-Sidi et al., 2018*; *Tejeda-Muñoz et al., 2019*). Wnt-induced macropinocytosis also causes the internalization of focal adhesions (FAs) and integrins from the cell surface into MVBs (*Tejeda-Muñoz et al., 2022c*). The requirement of macropinocytosis for Wnt signaling is revealed, for example, by the decrease in nuclear β-catenin when colorectal cancer (CRC) cells mutant for APC were treated with macropinocytosis inhibitors such as the derivative of the diuretic Amiloride known as 5-(*N*-Ethyl-*N*-isopropyl) Amiloride (EIPA) (*Tejeda-Muñoz et al., 2019*; *Tejeda-Muñoz and De Robertis, 2022a*; *Tejeda-Muñoz and De Robertis, 2022b*).

Macropinocytosis is emerging in the field as a promising target for cancer treatment (*Lambies and Commisso, 2022*). In addition, it has been found that the macropinocytosis induced by Wnt signaling causes significant lysosome acidification and activation (*Albrecht et al., 2020*). There is also accumulating evidence showing that the size and number of lysosomes increase during carcinogenesis (*Cardone et al., 2005*; *Kroemer and Jäättelä, 2005*; *Zhitomirsky and Assaraf, 2016*). Lysosomes, previously thought to be the garbage disposal of the cell, have emerged as central organelles in cancer (*Kirkegaard and Jäättelä, 2009*; *Kirkegaard and Jäättelä, 2009*).

From the considerations above, investigating the intersection of Wnt signaling, macropinocytosis, lysosomes, FAs, and membrane trafficking, is of great interest. While in most cells macropinocytosis requires stimulation of receptor tyrosine kinase (RTK) (*Haigler et al., 1979*; *Yoshida et al., 2018*), Ras (*Bar-Sagi and Feramisco, 1986*; *Commisso et al., 2013*), or Wnt (*Redelman-Sidi et al., 2018*; *Tejeda-Muñoz et al., 2019*), some cells, such as amoebae and macrophages, have constitutive macropinocytosis. Importantly, Swanson discovered that the phorbol ester PMA stimulated macropinocytosis even in macrophages that were thought to have fully activated constitutive macropinocytosis (*Swanson, 1989*).

In the present study, we asked whether phorbol ester could interact with canonical Wnt signaling. Using the *Xenopus* embryo, which provides a premier model system to analyze the Wnt pathway, we found that PMA, which lacked any effects on its own, greatly sensitized the embryo to dorsalization by Wnt signaling, leading to the formation of radial head structures lacking trunks. The dorsalizing effect of PMA on embryos was blocked by the macropinocytosis inhibitor EIPA, the lysosomal V-ATPase inhibitor Bafilomycin A (Baf), or the Rac1 inhibitor EHT1864. Rac1 is an upstream regulator of the p21-activated kinase 1 (Pak1) required for actin-mediated macropinocytosis. Lysosomal activity was specifically required at the 32-cell stage for the activity of the Wnt-mimicking GSK3 inhibitor Lithium Chloride (LiCl). Ectopic axes induced by dominant- negative GSK3 (DN-GSK3-β) or β-catenin mRNAs were blocked by EIPA or dominant-negative Rab7, indicating a widespread requirement of membrane trafficking for Wnt signaling.

The involvement of membrane trafficking in Wnt-driven cancer was supported by immunostaining results of arrayed human CRC histological sections, in which we found that the grade of malignancy increased Pak1, CD63 (an MVB marker), and V0a3 (a V-ATPase subunit that marks lysosomes) levels, while decreasing GSK3. In *Xenopus* embryos, twinning induced by Xwnt8 mRNA microinjection was blocked by inhibition of Rac1. In cultured CRC SW480 cells reconstituted with APC, the FA marker Tes was cytoplasmic, while the same cells mutant for APC (in which the Wnt pathway is activated) Tes was nuclear. The results suggest that macropinocytosis and membrane trafficking play an important role in embryogenesis and cancer progression.

## Results

### The phorbol ester PMA potentiates Wnt-like signals in *Xenopus* embryos via macropinocytosis and lysosomal acidification

Wnt is one of the earliest signals in the embryo and it induces the formation of the primary dorsal–ventral and head-to-tail axes (*Loh et al., 2016*; *Niehrs, 2012*). It is known that β-catenin is required for Wnt signaling activation on the dorsal side during early cleavage. Recently, we reported that inhibition of macropinocytosis with EIPA, or of lysosomal acidification with Baf, ventralized endogenous early dorsal axis formation, leading in particular to reduced head and brain structures (*Tejeda-Muñoz and De Robertis, 2022a*). Activation of lysosomes on the dorsal side of the embryo was detectable as early as the 64-cell stage (*Tejeda-Muñoz and De Robertis, 2022a*).

The tumor promoter PMA is known to increase macropinocytosis in macrophages (*Swanson, 1989*). To test its activity during early development, we used a sensitized system in which a small amount of the GSK3 inhibitor LiCl (4 nl at 300 mM in a single ventral injection at 4-cell) causes only a small dorsalization with slightly enlarged heads (*Kao et al., 1986*; *Tejeda-Muñoz and De Robertis, 2022a*). The dorsalizing effects of PMA were most striking when phenotypes were analyzed at the early tailbud stage (*Figure 1A–H*). LiCl alone expanded head structures only slightly and PMA alone was without effect (*Figure 1A, B, E*). However, the combination of PMA and LiCl lead to radially dorsalized embryos consisting of head structures lacking trunks (compare *Figure 1B–F*). This striking phenotype was eliminated by co-injection of EIPA or Baf, suggesting that macropinocytosis and lysosomal acidification are required for the effect of PMA (*Figure 1G, H*). Injection of Baf alone decreased head and neural development, supporting a requirement for lysosomes for endogenous dorsal axis formation (*Figure 1D*). Target genes of the early Wnt signal can be detected at blastula (stage 9.5, just after the start of zygotic transcription) by measuring transcripts of *Siamois* and *Xnr3* by qRT-PCR (*Tejeda-Muñoz and De Robertis, 2022a*). As shown in *Figure 1I*, PMA alone (4 nl at 500 nM) was without any effect, LiCl increased the Wnt target genes, and the combination of PMA and LiCl synergized significantly elevated *Siamois* and *Xnr3* transcript levels, while EIPA and Baf blocked the increase in *Siamois* and *Xnr3* caused by PMA/LiCl (*Figure 1I*).

Using the BAR (β-catenin activity reporter) assay system (*Biechele and Moon, 2008*) in a cultured HEK293 permanent BAR/Renilla cell line, PMA synergized with LiCl when assayed by transcriptional BAR luciferase activity (*Figure 1J*). PMA on its own had no detectable effect on BAR reporter activity (*Figure 1—figure supplement 1J*). In this cell line, PMA cooperated with LiCl to increase β-catenin levels (*Figure 1—figure supplement 1A–H*). PMA alone caused a slight increase in β-catenin protein staining, but this lacked statistical significance (*Figure 1—figure supplement 1C, I*), presumably due to a small amount of endogenous Wnt secreted by this cell line (*Colozza et al., 2020*). Importantly, the effect of PMA on the activation of the Wnt pathway by GSK3 inhibition was abrogated by the inhibition of V-ATPase with Baf (*Figure 1—figure supplement 1H, I*).

To test the effect of PMA on Wnt-induced macropinocytosis we used an uptake assay in which 3T3 cells were treated with dextran for 1 hr (*Figure 1K–P*). The macropinocytosis marker TMR-dextran 70 kDa has a hydrated diameter of more than 200 nM and is diagnostic of large endocytic cups, in particular when sensitive to the macropinocytosis inhibitor EIPA (*Commisso et al., 2013*; *Commisso et al., 2014*). In the presence of Wnt3a, but not in the absence, PMA strongly induced dextran uptake and this effect was blocked by EIPA (*Figure 1K–P*, quantitated in *Figure 1Q*). PMA also increased dextran uptake in the SW480 CRC cell line that has constitutive Wnt signaling, even in the absence of added Wnt (*Figure 1—figure supplement 1J, K*). In time-lapse video microscopy of SW480 cells with the plasma membrane tagged by myristoylated-GFP (mGFP), PMA addition triggered the abundant

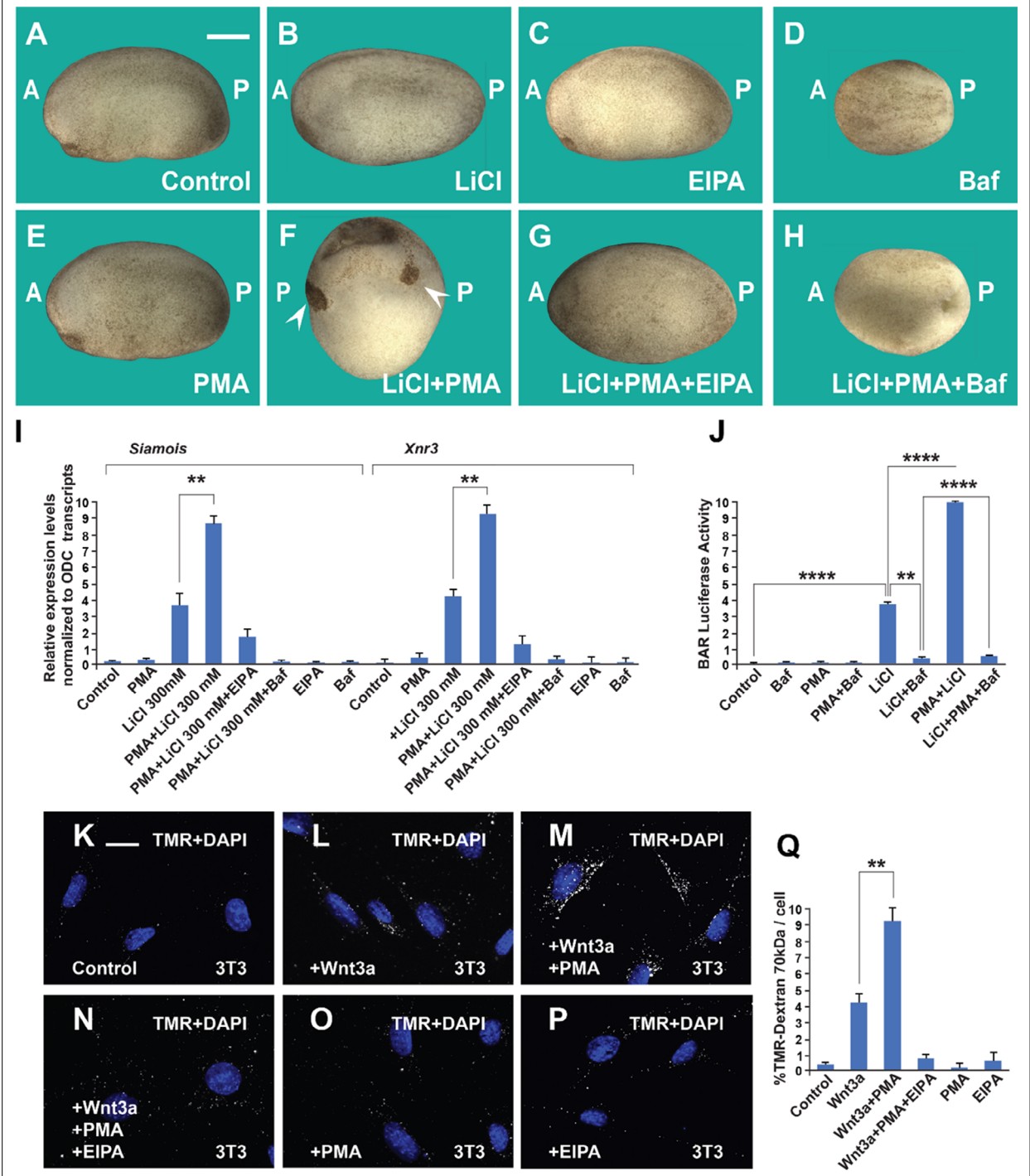

**Figure 1.** The phorbol ester phorbol 12-myristate 13-acetate (PMA) synergizes with GSK3 inhibition and Wnt3a and this cooperation requires macropinocytosis and lysosome acidification. (**A**) Uninjected control embryo at stage 24. (**B**) Lithium Chloride (LiCl) (4 nl 300 mM, 1x ventral) dorsalized the embryo only slightly. (**C**) 5-(*N*-Ethyl-*N*-isopropyl) Amiloride (EIPA) (1 mM, 1x ventral) alone did not produce a distinctive phenotype at this concentration. (**D**) The vacuolar ATPase (V-ATPase) inhibitor Bafilomycin A1 (Baf) (incubation with 5 µM for 7 min at 32 cells) ventralized embryos. (**E**) A single injection of PMA (500 nM) did not show any phenotypic effect. (**F**) PMA and LiCl strongly synergized, resulting in embryos with radial heads. (**G**) Co-injection of the EIPA macropinocytosis inhibitor blocked dorsalization caused by LiCl plus PMA. (**H**) Incubation with Baf lysosomal acidification inhibitor blocked dorsalization caused by LiCl plus PMA. Number of embryos was as follows: A = 70, 100%; B = 80; 98%; C = 75, 94%; D = 82, 92%; E = 76, 100%; F = 112, 98%; G = 75, 93%; H = 85, 98%. Scale bar, 500 µm. (**I**) PMA increases the effects of LiCl in Wnt signaling in early embryos. qRT-PCR analysis at blastula stage 9.5 of the Wnt target genes Siamois and Xnr3 normalized for Ornithine decarboxylase (ODC) transcripts. Embryos received a single injection of 4 nl of LiCl 300 mM with or without PMA (500 nM). Error bars denote standard error of the mean (SEM) ($n \geq 3$) (**p < 0.01).

*Figure 1 continued on next page*

Figure 1 continued

(**J**) Luciferase assay in HEK293BR cells showing cooperation between PMA and LiCl, and its inhibition by Baf. (**K–P**) PMA and Wnt3a cooperated in stimulating macropinocytosis, which was sensitive to inhibition by EIPA, in 3T3 cells. TMR-dextran 70 kDa was added for 1 hr. (**Q**) Quantification of the Wnt3a + PMA results. Experiments with cultured cells were biological triplicates. Error bars denote standard deviation (****p < 0.0001 and **p < 0.01).

The online version of this article includes the following figure supplement(s) for figure 1:

**Figure supplement 1.** In cultured cells, the phorbol ester phorbol 12-myristate 13-acetate (PMA) cooperates with GSK3 inhibition in β-catenin signaling and increases dextran macropinocytosis.

formation of macropinosome cups within minutes of addition when compared to DMSO (Dimethyl-sulfoxide) controls (**Video 1**).

These results in embryos and cultured cells indicate that the tumor promoter PMA potentiates Wnt signaling and that this cooperation requires macropinocytosis and lysosomal acidification.

## Lysosomal activity is required at the 32-cell stage for early Wnt activation

In previous work, we had shown that immersion of embryos in the V-ATPase inhibitor Baf at 5 µM for only 7 min at the 32-cell stage resulted in ventralized microcephalic embryos lacking cement glands (*Tejeda-Muñoz and De Robertis, 2022a*). To test whether this ventralization is caused by inhibition of the endogenous Wnt pathway signal, we now carried out order-of-addition experiments. When an initial Baf incubation of 7 min was followed by a second brief incubation in LiCl while still at the 32-cell stage, the ventralization by Baf was rescued, and the resulting tailbud embryos had a LiCl-like dorsalized phenotype (*Figure 2A–D*). The initial effect of GSK3 by LiCl is to trigger macropinocytosis (*Albrecht et al., 2020*). The phenotypes are supported by the activation of Wnt signaling at the blastula stage, as indicated by qRT-PCR of direct Wnt target genes (*Figure 2E*). In the converse experiment, when embryos were first briefly treated with LiCl, washed, and subsequently incubated for 7 min in Baf, the effect of lysosomal inhibition was dominant, resulting in ventralized tailbud embryos (*Figure 2F–J*). These order-of-addition experiments strongly support the proposal that membrane trafficking and endogenous lysosomal activity are required to generate the initial endogenous Wnt signal that takes place at the 32-cell stage of development (*Tejeda-Muñoz and De Robertis, 2022a*).

## Wnt pathway signaling in microinjected embryos or APC mutant cells requires macropinocytosis and membrane trafficking

Next, we tested the requirement for macropinocytosis of different components of the Wnt pathway using the axis duplication assay (*McMahon and Moon, 1989*). Microinjection of Wnt-mimic dominant-negative GSK3 (DN-GSK3-β) mRNA, a catalytically inactive form of GSK3, induced complete axis duplications, which were blocked by the co-injection of the macropinocytosis inhibitor EIPA (1 mM, 4 nl) (*Figure 3A–D*). Secondary axis induction by β-catenin mRNA was also abrogated by EIPA, as well as the membrane trafficking inhibitor dominant-negative Rab7 (DN-Rab7) (*Figure 3E–H*). β-Catenin is a transcriptional activator, and one might have not expected a requirement for membrane trafficking. However, axis formation is a multistep process and appears to require macropinocytosis and Rab7 activity in addition to transcriptional activation of Wnt target genes.

Macropinocytosis is an actin-driven process that is orchestrated by Pak1 kinase, which in turn is activated by Rac1. Rac1 activity is essential for the formation of lamellipodia and macropinocytic cups (*Hall, 1998*). EHT1864 (EHT) is a specific

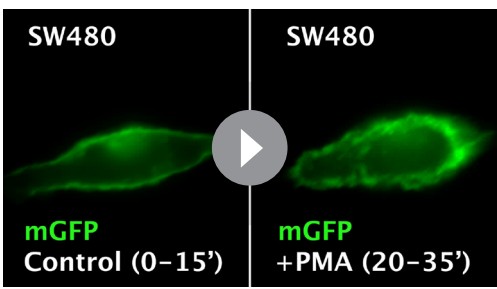

**Video 1.** An SW480 colorectal cancer (CRC) cell before and after treatment with phorbol 12-myristate 13-acetate (PMA) (0.3 µM); the plasma membrane was marked by transfection of myristoylated-GFP (mGFP) and filmed for 15 min. Note that PMA greatly enhanced plasma membrane vesicular activity typical of macropinocytosis. Control cell was treated with DMSO alone and shows low but detectable macropinocytosis cup formation. The arrowheads indicate two prominent macropinocytosis vesicles.

https://elifesciences.org/articles/89141/figures#video1

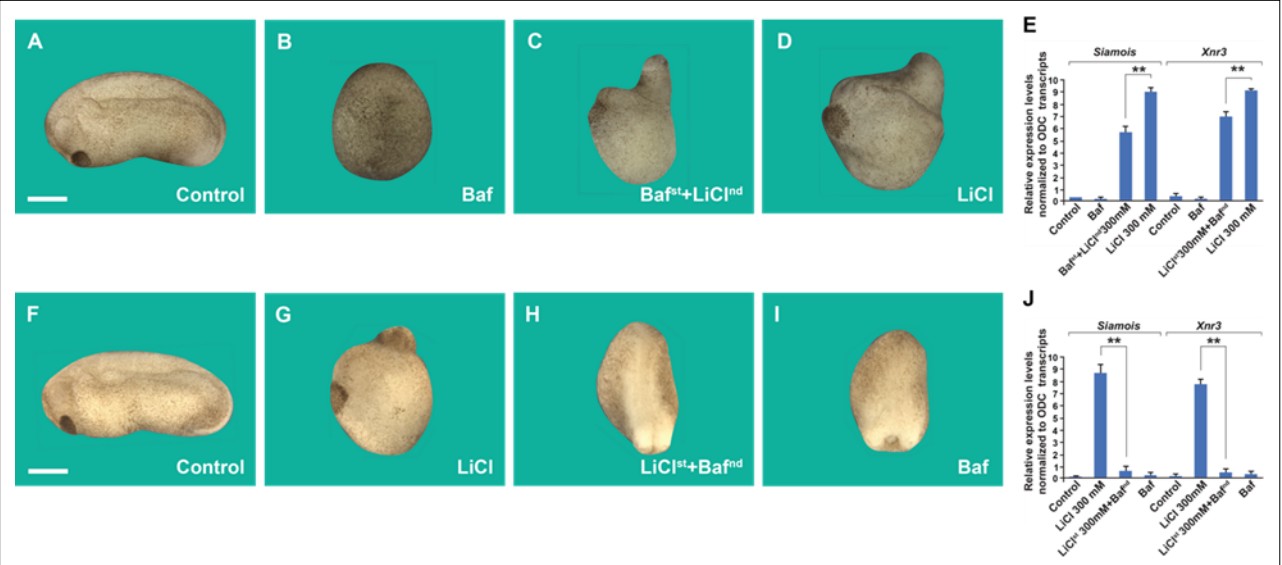

**Figure 2.** Order-of-addition experiment showing that lysosomal acidification at 32-cell stage is required for early Wnt/β-catenin activation. (**A**) Untreated embryo. (**B**) Baf treatment (5 µM for 7 min) inhibited endogenous axis formation resulting in ventralized embryos lacking heads at early tailbud. (**C**) Embryos treated with Baf, washed, and then immersed in 300 mM Lithium Chloride (LiCl) an additional 7 min formed heads, resulting in dorsalized embryos with enlarged heads and short trunks. Note that all treatments were done at 32-cell stage, which is critical for the early Wnt signal. (**D**) LiCl treatment alone dorsalized embryos. (**E**) Quantitative RT-PCR (qPCR) for Wnt target genes *Siamois* and *Xnr3* at blastula, confirming that the phenotypic effects are due to early activation of the Wnt pathway. (**F**) Untreated embryo. (**G**) Embryos incubated with LiCl at 32-cell were dorsalized. (**H**) Embryos first treated with LiCl and then subsequently with Baf resulted in ventralized embryos lacking heads. (**I**) Baf treatment caused ventralization and small heads. (**J**) Quantitative RT-PCR for Wnt target genes *Siamois* and *Xnr3* showing that inhibiting V-ATPAse at the 32-cell stage blocks the effect of earlier LiCl treatment. The numbers of embryos analyzed were as follows: A = 11 5, 100%; B = 124, 98% with phenotype; C = 132, 97%; D = 99%; F = 132, 100%; G = 135, 99%; H = 126, 97%; I = 129, 98% (scale bars, 500 µm.). Error bars denote standard error of the mean (SEM) ($n \geq 3$) (**$p < 0.01$).

inhibitor of Rac1 that has the practical advantage of being water soluble (*Hampsch et al., 2017*). When embryos were immersed in EHT at the critical 32-cell stage for 7 min (10 mM), head structures were greatly reduced, and ventral posterior structures expanded (*Figure 4A, B*). This was accompanied by a significant increase of transcripts of the ventral markers Sizzled and Vent1 at the blastula stage (*Figure 4C*).

Microinjection of a small amount of EHT (1 mM, 4 nl, 1x ventral at 4-cell) was without effect on its own but was able to block the axis duplication effect of microinjecting Wnt8 mRNA (*Figure 4D–G*). Quantitative RT-PCR analyses at the blastula stage confirmed that inhibiting Rac1 strongly reduced the induction of the Wnt target genes Siamois and Xnr3 caused by xWnt8 mRNA microinjection at the late blastula stage (*Figure 4H*).

In SW480 CRC cells, which have constitutively activated Wnt due to mutation of APC (*Leibovitz et al., 1976*; *Faux et al., 2004*), EHT inhibited the size of cell spheroids in hanging drop cultures (*Figure 4—figure supplement 1A–C*) and, importantly, significantly inhibited macropinocytosis uptake of TMR-dextran in these cancer cells (*Figure 4—figure supplement 1D–F*).

Rac1 itself was stabilized in 3T3 cells by Wnt signaling, as indicated by experiments showing (1) that LiCl treatment increased levels of Rac1 protein (*Figure 4I–J"*), (2) that cells transfected with constitutively active β-catenin-GFP had higher levels of Rac1 than control untransfected cells (*Figure 4K–K"*), and (3) that Rac1 immunostaining was stabilized in cells transfected with CA-Lrp6-GFP when compared to untransfected cells (*Figure 4L–L"*).

The results show that signaling by the Wnt/β-catenin pathway requires macropinocytosis, Rac1, and membrane trafficking in *Xenopus* embryos and cancer cells.

## Tumor progression in human CRCs correlates with a role for membrane trafficking in Wnt/β-catenin signaling

CRC is particularly favorable for the analysis of tumor progression because it gradually accumulates mutations, and in over 85% of the cases the initial driver mutation is in the APC tumor suppressor

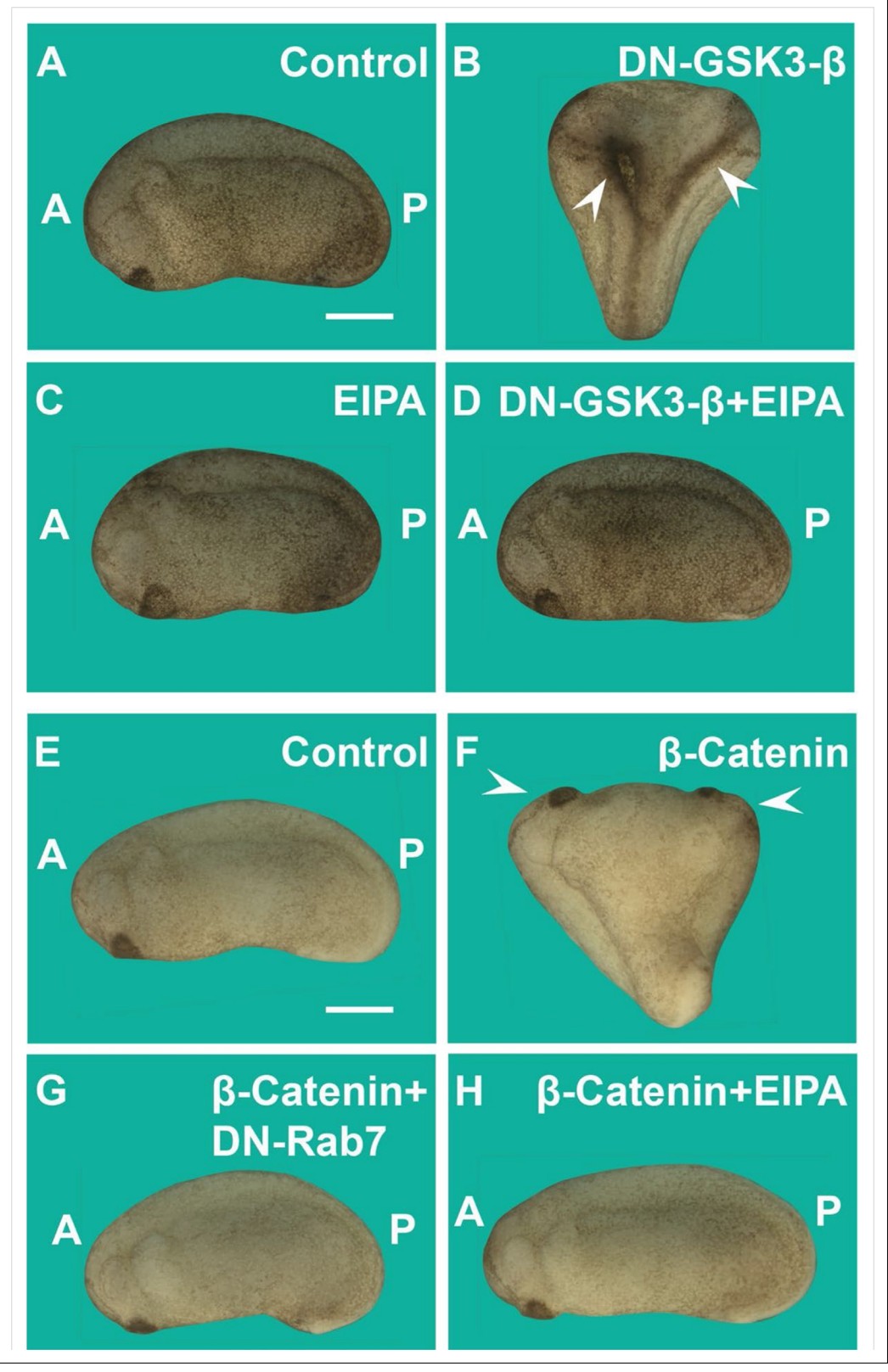

**Figure 3.** Induction of secondary axes by the GSK3/β-catenin pathway requires membrane trafficking. (**A**) Control embryo at early tailbud. (**B**) Embryo injected with DN-GSK3 (dominant-negative GSK3-β, 150 pg 1x ventral) mRNA, showing double axes (arrowheads). (**C**) Injection of 5-(*N*-Ethyl-*N*-isopropyl) Amiloride (EIPA) 1x ventral alone showed no phenotypic effect (4 nl,1 mM). (**D**) Co-injection of DN-GSK3 and EIPA blocked double axis formation.

*Figure 3 continued on next page*

*Figure 3 continued*

(**E**) Uninjected control embryo. (**F**) Activation of Wnt signaling via injection of β-catenin mRNA (80 pg) induced complete twinned axes (arrowheads). (**G**) Embryo co-injected with β-catenin mRNA and DN-Rab7 (500 pg) showing that membrane trafficking is required for secondary axis formation. (**H**) Co-injection of β-catenin mRNA and the macropinocytosis inhibitor EIPA blocks axial duplication. The numbers of embryos analyzed were as follows: A = 62, 100%; B = 75, 94% with double axes; C = 76, 100%; D = 70, 98%; E = 70, 100%; F = 73, 98%; G = 69, 75%; H = 82, 97%; four independent experiments (scale bars, 500 µm).

(***Kinzler and Vogelstein, 1996***; ***Weinberg and Weinberg, 2007***). This offered an opportunity to test our proposal that membrane trafficking and its molecular components play an important role in Wnt-driven cancers (***Albrecht et al., 2021***). Commercially available arrays of paraffin-sectioned histological sections containing 90 cases of adenocarcinoma of various grades I–IV and 90 samples of corresponding adjacent normal tissues (from TissueArray) were used for immunostaining. Arrays were double stained for β-catenin and antibodies against proteins involved in the macropinocytosis/MVB/lysosome/GSK3 pathway. Each individual section was evaluated, and normal colon, adenocarcinoma I, and adenocarcinoma IV were compared to assess the effects of cancer progression.

As shown in ***Figure 5***, β-catenin levels consistently correlated with grade IV malignancy when compared to normal colon or grade I cancers. Pak1, a kinase required for macropinocytosis and Wnt signaling (***Redelman-Sidi et al., 2018***; ***Albrecht et al., 2020***) was significantly elevated with increased malignancy (***Figure 5A–C"***). CD63, a tetraspan protein that marks MVB intraluminal vesicles (***Escola et al., 1998***), reached high levels in adenocarcinoma IV (***Figure 5D–F"***), particularly in cells with the highest β-catenin expression (compare ***Figure 5F–F'***). This increase in MVBs is in agreement with the GSK3 sequestration model of Wnt signaling (***Taelman et al., 2010***). The V-ATPase subunit V0a3, a marker for acidic lysosomes (***Ramirez et al., 2019***), also increased strongly with the CRC malignancy (compare ***Figure 5G–I***). Importantly, the cellular levels of total GSK3 decreased with malignancy (***Figure 5J–L"***, compare ***Figure 5J–L***), as predicted by the sequestration model (***Taelman et al., 2010***). During Wnt signaling, GSK3 is sequestered into MVBs which are trafficked into lysosomes and a decrease was to be expected after sustained Wnt activation. Previously, we had been unable to demonstrate a decrease in total cellular GSK3 levels during acute Wnt signaling experiments (***Taelman et al., 2010***). The co-localization of β-catenin with Pak1, CD63, V0a3, or GSK3 in human colorectal adenocarcinomas was quantified in ***Figure 5—figure supplement 1A–D***.

These results were confirmed by examining immunostaining levels of Pak1, V0a3, and GSK3 compared to β-catenin protein in mouse xenografts in a CD1 NU/NU nude mouse model injected with SW480 human CRC cells (in which Wnt activation is known to be caused by APC mutation, ***Faux et al., 2004***). In this case, tumor cells were compared to normal colon mouse tissue sections. The results showed that CRC cells had high levels of β-catenin, Pak1, and lysosomes (***Figure 5—figure supplement 1E–J***), while GSK3 levels were reduced (compare ***Figure 5—figure supplement 1K–L***) in agreement with the results in human tissue arrays.

These in vivo observations in human tumors are consistent with the view that macropinocytosis components, MVBs, and lysosomes increase with CRC malignancy. Further, the decrease in total GSK3 levels suggests a role in the trafficking and degradation of this key enzyme in cancer.

## Wnt affects the subcellular localization of FA components

We have previously reported a crosstalk between the Wnt and FA signaling pathways in which Wnt3a treatment rapidly led to the endocytosis of Integrin β1 and of multiple FA proteins into MVBs (***Tejeda-Muñoz et al., 2022c***). FAs link the actin cytoskeleton with the extracellular matrix (***Figure 6A***), and we now investigated whether FA activity is affected by Wnt signaling, PMA treatment, or CRC progression. FA components containing LIM domains can shuttle between the nucleus and cytoplasm (***Nix and Beckerle, 1997***; ***Kadrmas and Beckerle, 2004***; ***Anderson et al., 2021***). We examined whether constitutive Wnt signaling regulates FA trafficking using the SW480 and SW480APC system, in which the stable reconstitution with full-length APC restores these CRC cells to a non-malignant phenotype (***Faux et al., 2004***), which is accompanied by a decrease in β-catenin levels (***Figure 6B***). As shown in ***Figure 6C–D"***, the PET-LIM domain FA protein Tes (also known as Testin) (***Garvalov et al., 2003***) was cytoplasmic in SW480APC, but was nuclear in SW480 cells in which Wnt is activated by APC mutation. The nuclear localization of Tes in SW480 cells required

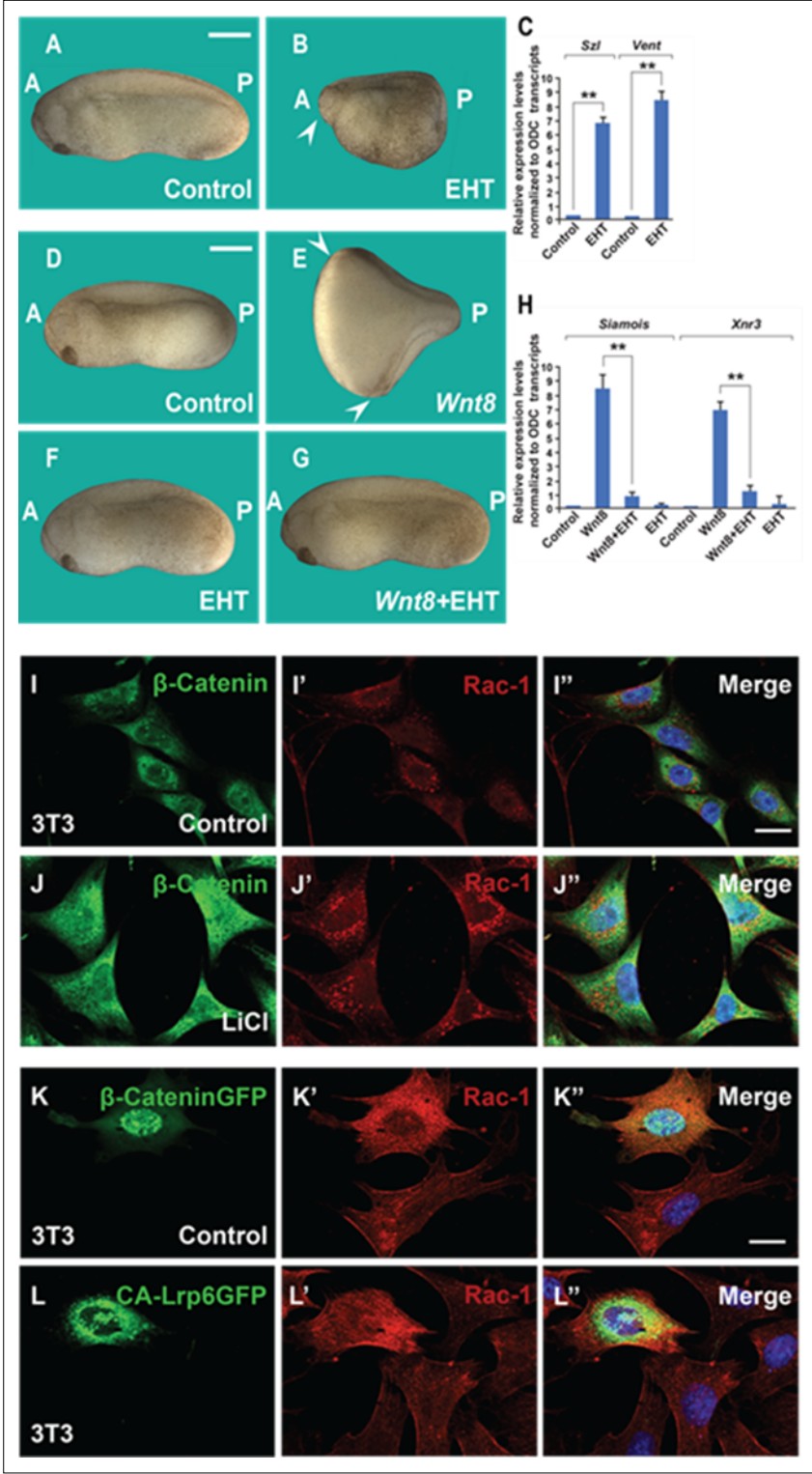

**Figure 4.** The Rac1 inhibitor EHT1864 blocks Wnt signaling in *Xenopus* embryos, and Rac1 levels are stabilized by treatments that increase Wnt/β-catenin signaling. (**A**) Uninjected control embryo. (**B**) Incubation of the *Xenopus* embryos with the Rac1 inhibitor EHT at 32-cell stage (7 min, 10 mM) resulted in a ventralized phenotype with a small head in the anterior (A, arrowhead) and expanded ventral structures in the posterior (P). Rac1 activity is required for macropinocytosis, see *Figure 4—figure supplement 1*. (**C**) qRT-PCR of gastrula stage embryos showing increased ventral markers Szl and Vent1 after Rac1 inhibition. (**D**) Control embryo. (**E**) Injection of Wnt8. mRNA (2 pg) induces complete duplicated axes (arrows). (**F**) Injected embryos with EHT (1 mM, 4 nl 1x ventral)

*Figure 4 continued on next page*

*Figure 4 continued*

alone showed no phenotypic effect at this concentration. (**G**) EHT co-injected with Wnt8 mRNA blocked double axis formation. (**H**) qRT-PCR of Wnt target genes *Siamois* and *Xnr3* at blastula confirming that Rac1 is required for early Wnt signaling. (**I–J″**) Treatment of 3T3 cells with 40 mM Lithium Chloride (LiCl) increases β-catenin and Rac1 levels. (**K–L″**) Transfection of 3T3 cells with stabilized constitutively active forms of β-cateninGFP or Lrp6GFP increased levels of Rac1 protein in transfected cells compared to untransfected ones. The numbers of embryos analyzed were as follows: A = 52, 100%; B = 47, 95% with ventralized small head phenotype; D = 58, 100%; E = 67; 97%; F = 64, 96%; G = 62, 97%, four independent experiments. Scale bars for embryos 500 μm; scale bars for immunofluorescence, 10 μm. qRT-PCR experiments represent biological replicates. Error bars denote standard error of the mean (SEM) ($n \geq 3$) (**$p < 0.01$).

The online version of this article includes the following figure supplement(s) for figure 4:

**Figure supplement 1.** Macropinocytosis is inhibited in SW480 colorectal cancer (CRC) spheroids by treatment with the Rac1 inhibitor EHT.

---

macropinocytosis, as suggested by treatment with EIPA which resulted in its re-localization to the cytoplasm (*Figure 6E–F″*). In the nude mouse xenograft model, Tes was abundant in the cytoplasm of normal colon epielium, while in transplanted SW480 cells it colocalized with β-catenin in the nucleus of those cells with the highest β-catenin levels (*Figure 6G–I″*). The function of TES is poorly understood, but it has been proposed to act as a tumor suppressor (*Tatarelli et al., 2000*; *Tobias et al., 2001*).

Focal adhesion kinase (FAK) is a key regulator of FA maintenance and signaling, known to interact with the Wnt pathway in various ways (*Chuang et al., 2022*). In HEK293 BAR/Renilla cells, treatment with the FAK inhibitor PF-00562271 strongly decreased β-catenin signaling by LiCl inhibition of GSK3 (*Figure 6J*; see also *Figure 6—figure supplement 1A–B″*). In the context on the *Xenopus* embryo, treatment with the FAK inhibitor (100 μM, 7 min at 32-cell stage) resulted in ventralized embryos with reduced dorsal-anterior structures such as central nervous system, in particular brain structures stained by the pan-neural marker Sox2 (*Figure 6K′–L″*). The addition of the tumor promoter PMA to SW480 cells (in which the Wnt pathway is constitutively activated) strongly increased protein levels of β-catenin and FAK protein (*Figure 6—figure supplement 1C–D″*). The effect of PMA on β-catenin and FAK levels was abrogated by EIPA or Baf treatment (*Figure 6—figure supplement 1C–H″*), suggesting that it requires macropinocytosis and lysosome activity.

Taken together, the results suggest a possible involvement of FA components and FAK in Wnt signaling, in particular during the endogenous early Wnt signal in *Xenopus* embryos. The results also suggest that nuclear Tes could serve as a Wnt-regulated marker in CRC. The regulated shuttling of FA components between nucleus and cytoplasm is probably just the tip of the iceberg of the role of membrane trafficking in Wnt-driven cancers.

## Discussion

The present work was inspired by the discovery that PMA increases macropinocytosis (*Swanson, 1989*). PMA plays an important role in our understanding of tumor biology because it is the archetypal tumor promoter, able to promote tumor growth without further mutation of the DNA (*Berenblum, 1941*; *Weinberg and Weinberg, 2007*). Our experiments support the role of membrane trafficking in canonical Wnt signaling in embryos and cancer cells and are summarized in the model in *Figure 7*. The main finding was a synergistic positive effect between the tumor promoter phorbol ester (PMA) and activation of the Wnt/β-catenin pathway that was blocked by EIPA or Baf (*Figure 1*). EIPA is a macropinocytosis inhibitor of the $Na^+/H^+$ exchanger at the plasma membrane; it causes acidification of the submembranous cytoplasm and prevents actin polymerization (*Koivusalo et al., 2010*). Bafilomycin A (Baf) is an inhibitor of the V-ATPase that drives membrane trafficking and lysosomal acidification. PMA activates the PKC pathway (*Takai et al., 1979*; *Nishizuka, 1984*), and there is previous literature on the crosstalk between the PKC and Wnt pathways (*Schwarz et al., 2013*; *Goode et al., 1992*). What is novel in the present paper is the crosstalk between macropinocytosis, membrane trafficking, lysosomes, FAs, PMA, and Wnt signaling.

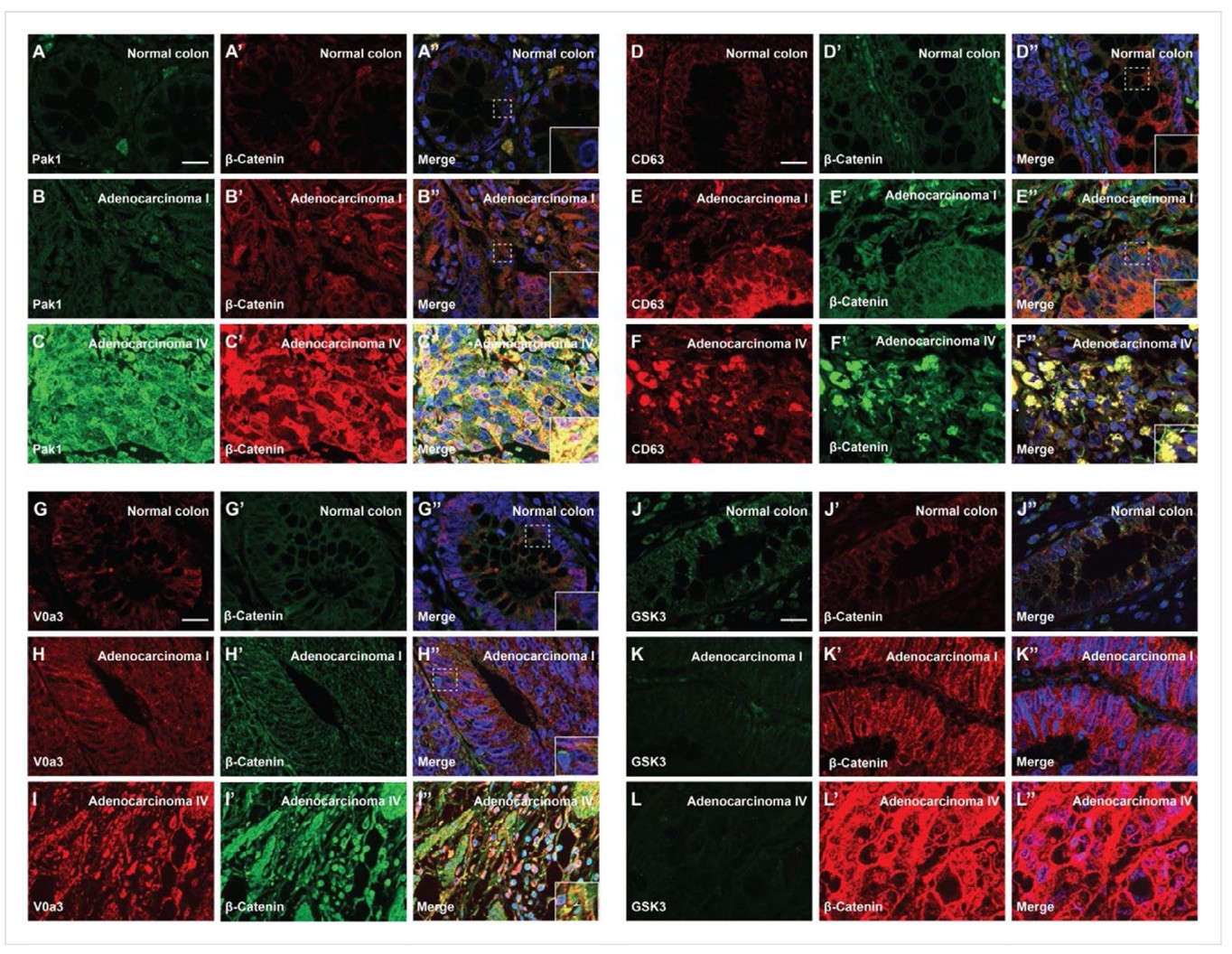

**Figure 5.** Degree of human colorectal cancer (CRC) malignancy is associated with increased macropinocytosis/multivesicular body (MVB)/lysosome markers and decreased GSK3 levels. (**A–A'**) Normal human colon paraffin section stained with the macropinocytosis marker Pak1 and β-catenin, respectively.(**A"**) Merged image with DAPI (4',6-diamidino-2-phenylindole), a few cells colocalize both markers. (**B–B"**) Pak1 and β-catenin levels are moderately increased at an early stage I CRC adenocarcinoma. (**C'–C"**) Strong colocalization between Pak1 and β-catenin was observed in advanced stage IV CRC (inset). (**D–D"**) Normal human colon section stained with the MVB marker CD63 and β-catenin; colocalization was not observed. (**E–E"**) CD63 and β-catenin were stabilized in adenocarcinoma I, and moderate colocalization was found between CD63 and β-catenin. (**F–F'**) CD63 and β-catenin were strongly stabilized and colocalized in adenocarcinoma stage IV CRC. (**F"**) Merge; note the striking colocalization between the MVB marker CD63 and β-catenin in advanced stages of cancer (inset). (**G–G"**) Normal colon stained for V0a3 (a subunit of V-ATPase that marks lysosomes) and β-catenin. (**H–H"**) Stage I adenocarcinoma with moderately increased levels of lysosomes and β-catenin. (**I–I"**) Strong co-localization of lysosomes and β-catenin in stage IV CRC (see inset). (**J–J"**) Human colon array stained with the GSK3 and β-catenin. (**K–K"**) GSK3 decreases, and β-catenin increases, at early stages of carcinogenesis, (**L–L"**) GSK3 levels are very low in advanced CRC compared to normal human colon, while β-catenin levels are very high. Scale bars, 10 µm. Also see *Figure 5—figure supplement 1* for quantifications and mouse xenografts of CRC cell.

The online version of this article includes the following figure supplement(s) for figure 5:

**Figure supplement 1.** Malignancy of colorectal cancer positively correlates with macropinocytosis, multivesicular body (MVB), and lysosome markers, and inversely correlates with GSK3 levels.

## Macropinocytosis, Rac1, and Wnt signaling

Wnt signaling triggers macropinocytosis and lysosomal acidification (*Albrecht et al., 2020*). In cultured cells, Wnt3a-induced macropinocytosis was increased by PMA and blocked by EIPA. In embryo microinjections, Baf blocked the synergy between PMA and the Wnt-mimic LiCl. Baf also blocked development of the endogenous axis, particularly the head region when embryos were briefly immersed in Baf

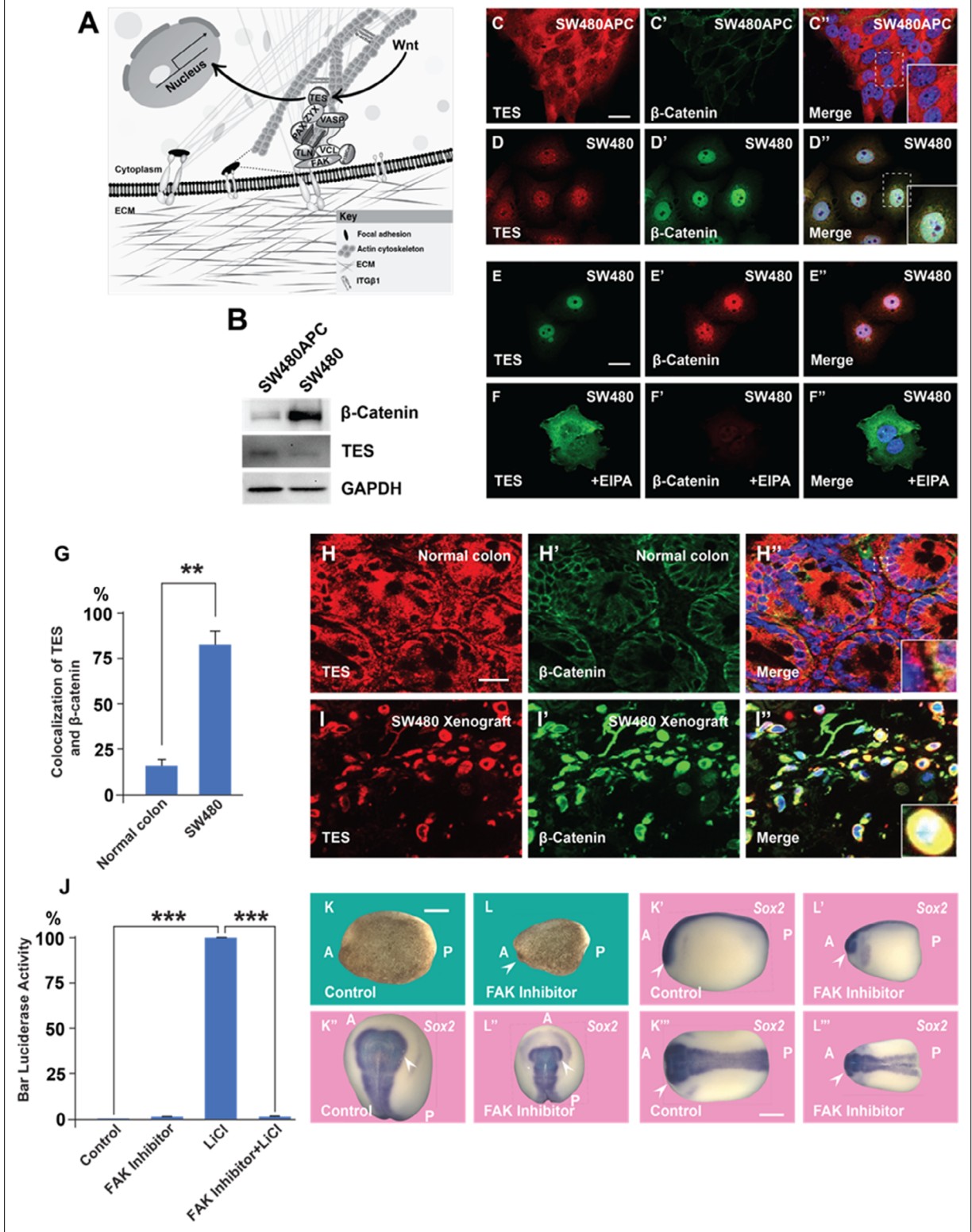

**Figure 6.** A focal adhesion protein changes its nucleocytoplasmic distribution after Wnt signaling activation, and focal adhesion kinase inhibition affects Wnt/β signaling. (**A**) Diagram showing how focal adhesions connect the actin cytoskeleton to the extracellular matrix via integrins; Wnt signaling changes the distribution of the focal adhesion protein Tes from focal adhesion sites to the nucleus. (**B**) Western blot of SW480 cells with and without stable expression of Adenomatous Polyposis Coli (APC); note the reduction in Tes expression and increase in β-catenin levels in SW480 cells lacking full-length APC. GADPH was used as loading control. (**C–C'**) Colon cancer cell line SW480 stably restored with full-length APC stained with Tes and

*Figure 6 continued on next page*

*Figure 6 continued*

β-catenin antibodies. Note that in SW480APC cells TES is absent from the nucleus (inset). (**D–D″**) In SW480 colorectal cancer (CRC) cells both Tes and β-catenin are located the nucleus (inset). (**E–E″**) Colon cancer cell line SW480 stained with Tes (in green) and β-catenin (in red) antibodies as a control. (**F–F″**) Inhibiting macropinocytosis with 5-(*N*-Ethyl-*N*-isopropyl) Amiloride (EIPA) (40 μM) treatment restored the cytoplasmic localization of the focal adhesion protein Tes in SW480 cells and strongly inhibited β-catenin levels. (**G**) Quantification of colocalization between Tes and β-catenin in normal mouse colon and SW480 xenografts. Error bars denote standard error of the mean (SEM) (*n* ≥ 3) (**\*\*p < 0.01**) (**H–H′**) Immunohistochemistry of the CD1 NU/NU normal mouse colon showing Tes and β-catenin distribution. (**H″**) Merge showing modest co-localization (inset). (**I–I″**) Immunohistochemistry images of the CD1 NU/NU mouse xenograft model showing nuclear TES, which is strongly colocalized with β-catenin levels in transplanted human SW480 cancer cells (inset). Scale bars, 10 μm. (**J**) Reporter assay in HEK293T cells stably expressing β-catenin activity reporter (BAR) and Renilla, showing strong inhibition of the β-catenin transcriptional activity induced by Lithium Chloride (LiCl) (40 mM) by focal adhesion kinase (FAK) inhibitor PF-00562271 (20 μM) after overnight treatment. Error bars denote standard error of the mean (SEM) (*n* ≥ 3) (**\*\*\*p < 0.001**). (**K–L‴**) Immersion of *Xenopus* embryos in FAK inhibitor (7 min at 32-cell stage) inhibits head and dorsal development. Bright field and several angles of in situ hybridization with pan-neural marker Sox2 are shown to demonstrate reduction in brain structures indicated by the arrowhead; scale bar 500 μm.

The online version of this article includes the following source data and figure supplement(s) for figure 6:

**Source data 1.** Original files with the uncropped western blots used in *Figure 6B*.

**Figure supplement 1.** The phorbol ester phorbol 12-myristate 13-acetate (PMA) stabilizes focal adhesion kinase (FAK) and enhances β-catenin levels in cells with constitutive Wnt signaling.

at the 32-cell stage, as previously reported (*Tejeda-Muñoz and De Robertis, 2022a*). Using order-of-addition experiments in which embryos were treated with 7 min pulses of Baf or the GSK3 inhibitor LiCl, we showed that the requirement for lysosome acidification was due to inhibition of the Wnt pathway at the crucial 32-cell stage (*Figure 2*). Co-injection experiments of DN-GSK3 or β-catenin mRNAs with DN-Rab7 or EIPA supported the view that membrane trafficking and macropinocytosis are essential for the induction of secondary axes in *Xenopus* (*Figure 3*).

Rac1 is an upstream regulator of the Pak1 kinase required for the actin machinery that drives macropinocytosis (*Redelman-Sidi et al., 2018*). Rac1 is a member of the family of small Rho GTPases that drives the formation of lamellipodia, membrane ruffles, and macropinocytosis (*Hall, 1998*; *Egami et al., 2014*; *Fujii et al., 2013*). The Rac1 inhibitor EHT1864 (*Hampsch et al., 2017*) strongly ventralized the endogenous axis of embryos after brief incubation at the critical 32-cell stage (*Figure 4*). In co-injection experiments EHT blocked the axis-inducing effect of microinjected xWnt8 mRNA. Rac1 is a regulator of multiple signaling cascades and has been reported to increase the nuclear translocation of β-catenin (*Wu et al., 2008*). Other work has shown that Rac1 binds to β-catenin and promotes the formation of nuclear β-catenin/LEF1 complexes (*Jamieson et al., 2015*). In the context of the present study, Rac1 protein levels were increased by treatments that enhanced Wnt/β-catenin signaling. The results suggest that Rac1 may crosstalk with Wnt through macropinocytosis in early development.

## Lysosomes, cancer and FAs

The lysosome is considered a promising target for the treatment of cancer (*Fehrenbacher and Jäättelä, 2005*; *Lawrence and Zoncu, 2019*). We used human CRC tissue arrays to analyze whether lysosomes and membrane trafficking correlate with cancer progression. Most colorectal tumors are initiated by mutation of APC, which drives Wnt signaling (*Kinzler and Vogelstein, 1996*; *Segditsas and Tomlinson, 2006*). It was found that advanced grades of colon carcinoma correlated with increased immunostaining for Pak1 (a driver of macropinocytosis), CD63 (a MVB/late endosome marker), and V0a3 (a lysosomal V-ATPase marker) (*Figure 5*). Importantly, cancer malignancy correlated with low levels of GSK3 staining, both in human samples and nude mouse SW480 xenografts. Cancer cells require lysosome function and have changes in lysosomal volume and subcellular localization during oncogenic transformation (*Kirkegaard and Jäättelä, 2009*; *Kirkegaard and Jäättelä, 2009*; *Gocheva and Joyce, 2007*; *Kolwijck et al., 2010*). V-ATPase has been associated with acidification of the extracellular milieu in tumors (*Capecci and Forgac, 2013*; *Hinton et al., 2009*; *Perona and Serrano, 1988*). Extracellular tumor acidification is probably due to increased numbers of lysosomes which are exocytosed, since V0a3 was located within the cytoplasm in advanced cancer or xenografts in mice (*Figure 5I* and *Figure 5—figure supplement 1I*). The results reported here support the view (*Taelman et al., 2010*; *Tejeda-Muñoz and De Robertis, 2022a*) that Wnt-driven lysosomal acidification and GSK3 sequestration may play a role in cancer. In addition, MVB/lysosome components could serve as markers for advanced Wnt-driven cancers.

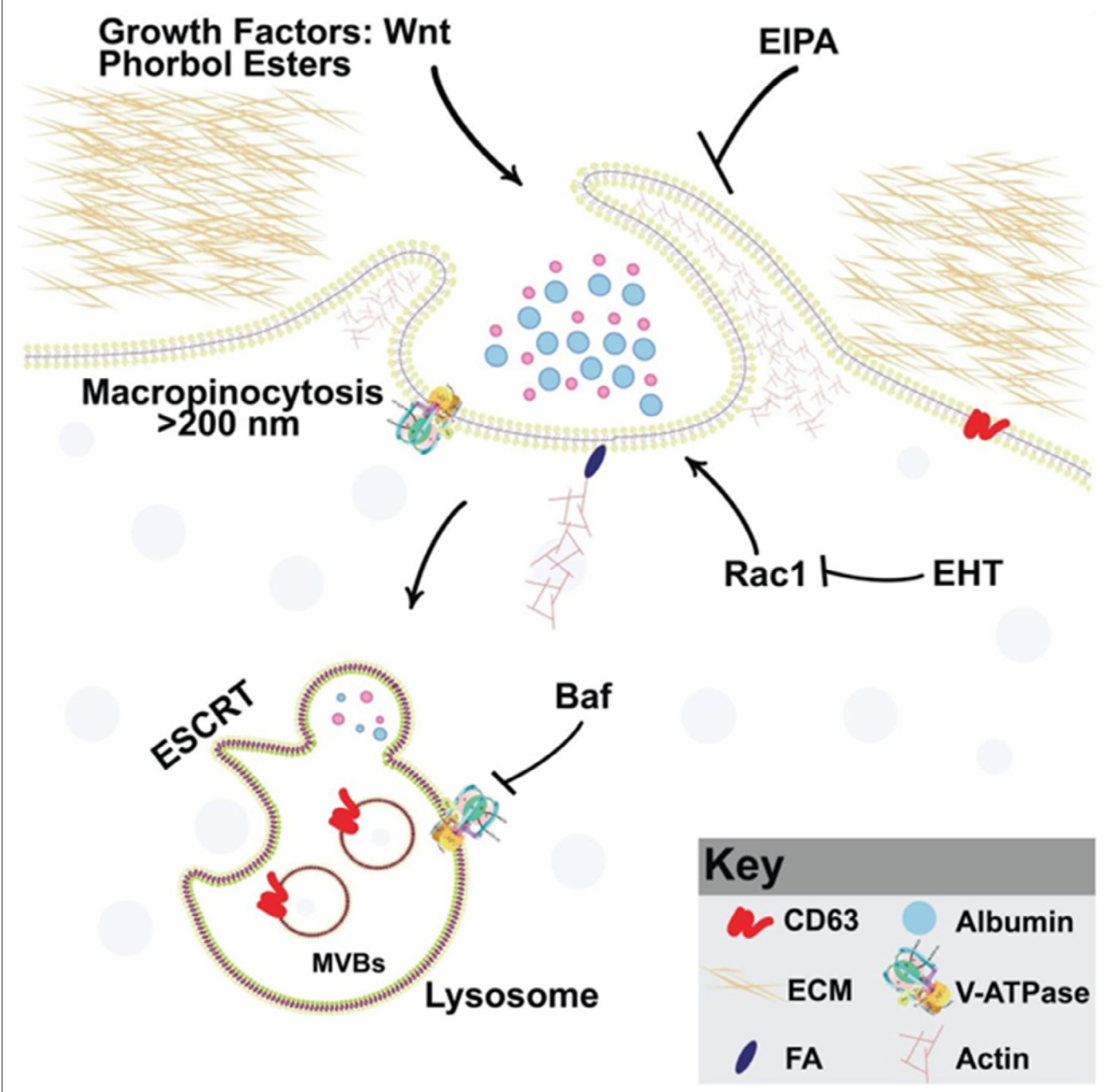

**Figure 7.** Model of phorbol 12-myristate 13-acetate (PMA) synergy with Wnt signaling through macropinocytosis and membrane trafficking Wnt and PMA are activators of macropinocytosis, which is a cell drinking process driven by an actin meshwork that requires the small GTPase Rac1. Membrane trafficking into multivesicular bodies (MVBs)/lysosomes (marked by CD63) requires acidification by V-ATPase, which is inhibited by Bafilomycin A (Baf). Phorbol esters such as PMA stimulate macropinocytosis and synergize with Wnt signaling. Inhibiting macropinocytosis with the membrane trafficking inhibitors 5-(*N*-Ethyl-*N*-isopropyl) Amiloride (EIPA), Baf, or the Rac1 inhibitor EHT1864 block Wnt signaling and its cooperation with the tumor promoter PMA.

The cytoskeleton interacts with proteins of the extracellular matrix though FAs (*Geiger et al., 2001*). We previously reported that Wnt causes the endocytosis of FA proteins and depletion of Integrin β1 from the cell surface (*Tejeda-Muñoz et al., 2022c*). We now report that Tes, a member of the PET-LIM family of FA proteins, changed subcellular localization after Wnt activation in SW480 cells. Other FA proteins shuttle between the nucleus, cytoplasm, and FA sites, and may play important roles in cancer

(*Chaturvedi et al., 2012*; *Nix and Beckerle, 1997*). FAK is a key signaling kinase that regulates FA signaling. In SW480 cells (which have constitutive Wnt signaling), FAK protein levels were increased by PMA, and this was inhibited by EIPA or Baf. The FAK inhibitor PF-00562271 inhibited β-catenin signaling in Luciferase assays, and reduced dorsal-anterior structures such as CNS in embryos after incubation for 7 min at 32-cell stage that is critical for the early Wnt signaling. These results suggest a possible crosstalk between FAs, Wnt signaling and tumor promoters in cancer progression and embryonic development.

## Cooperation between PMA and Wnt

PMA activates PKC signaling by mimicking its physiological second messenger DAG, an activator of PKC. There is previous literature implicating PKC isozymes in the regulation of the Wnt signaling pathway (*Fang et al., 2002*; *Goode et al., 1992*) and that different PKC isoforms can have positive or negative effects on Wnt/β-catenin signaling (*Ohno and Nishizuka, 2002*). However, the molecular role of PKC isoforms in the Wnt signaling pathway remains poorly understood (*Bhatia and Spiegelman, 2005*; *Schwarz et al., 2013*). It has been reported that PMA can stabilize CK1ε (Casein Kinase 1 epsilon), enhance its kinase activity, and induce LRP6 phosphorylation at Thr1479 and Ser1490, leading to the activation of the Wnt/β-catenin pathway (*Su et al., 2018*). In the context of the present study, it is relevant that Wnt signaling rapidly triggers macropinocytosis and trafficking into lysosomes (*Tejeda-Muñoz et al., 2019*). Macropinocytosis requires the activation of phosphoinositide 3 kinase (PI3K) which leads to the formation of patches of PIP3 in the inner plasma membrane leaflet that initiate the formation of macropinocytic cups (*Egami et al., 2014*; *Yoshida et al., 2018*). During macropinosome formation, the $PIP_3$ lipid is then converted into DAG by phospholipase C gamma 1 (PLCγ1), and DAG activates PKC signaling (*Yoshida et al., 2018*). It seems likely that PMA may facilitate Wnt/β-catenin signaling by mimicking the DAG lipid in macropinosomes. We have tested this hypothesis by direct addition of DAG to cells and found cooperation with Wnt signaling (*Azbazdar et al., 2023*).

In the future, it will be interesting to study the role of chronic inflammation on macropinocytosis and tumor promotion. Active PKC phosphorylates many targets, including IκB kinase (IKK), which becomes activated and phosphorylates inhibitor of NF-κB (IκB), targeting it for degradation and triggering inflammation (*Weinberg and Weinberg, 2007*). Tumor progression is promoted by many chronic inflammatory agents such as turpentine, distilled alcoholic beverages, hepatitis viruses, and *Helicobacter pylori* that do not cause mutations by themselves but regulate the expansion of previously mutant clones (*Rous and Kidd, 1941*; *Weinberg and Weinberg, 2007*). Taken together, the results presented here suggest that macropinocytosis, lysosomal activity, and membrane trafficking are possible therapeutic targets for tumor progression in Wnt-driven cancers.

## Materials and methods

**Key resources table**

| Reagent type (species) or resource | Designation | Source or reference | Identifiers | Additional information |
|---|---|---|---|---|
| Cell line (*Homo sapiens*) | HeLa (human cervical adenocarcinoma) | ATCC | RRID: CVCL_0030 | |
| Cell line (*Homo sapiens*) | SW480 (human colorectal) | ATCC | RRID:CVCL_0546 | |
| Biological sample (*Xenopus laevis* male) | Sperm | *Xenopus* I | 5215 Albino *Xenopus laevis* Sexually Mature Male/-/-/M | Used for fertilization |
| Biological sample (*Xenopus laevis* female) | Egg | *Xenopus* I | 4280 Sexually Mature Female 10.5–11 cm/-/-/F | Used for fertilization |
| Biological sample (CD1 NU/NU nude mouse female) | | Charles River Laboratories | | Xenograft |
| Antibody | Pak1 antibody rabbit polyclonal | Abcam | Cat# ab131522, RRID:AB_11156726 | IF (1:100) |
| Antibody | FAK antibody rabbit polyclonal | Cell Signaling | Cat# 3285, RRID:AB_2269034 | IF (1:100) |

*Continued on next page*

*Continued*

| Reagent type (species) or resource | Designation | Source or reference | Identifiers | Additional information |
|---|---|---|---|---|
| Antibody | Rac-1 antibody rabbit polyclonal | Thermo Fisher | Cat# PA1-091 | IF (1:100) |
| Antibody | CD63 antibody mouse monoclonal | Abcam | Cat# ab59479 RRID:AB_ 940915 | IF (1:100) |
| Antibody | TES antibody rabbit polyclonal | ATLAS | Cat# HPA018123 RRID:AB_1857900 | IF (1:100) |
| Antibody | GSK3 antibody mouse monoclonal | Abcam | Cat# ab93926, RRID:AB_10563643 | IF (1:100) |
| Antibody | Transferrin receptor (TfR) antibody mouse monoclonal | Thermo Fisher | Cat# 13-6800 | IF (1:100) |
| Antibody | ATP6V0a3 antibody rabbit polyclonal | Novus | Cat# nbp1-89333 RRID:AB_11016312 | IF (1:100) |
| Antibody | GAPDH antibody rabbit monoclonal | Cell Signaling | mAb #2118 | WB (1:1000) |
| Antibody | β-Catenin antibody rabbit polyclonal | Thermo Fisher Scientific | Cat# 71-2700; RRID:AB_ 2533982 | IF (1:100) |
| Antibody | β-Catenin antibody mouse monoclonal | Thermo Fisher Scientific | Cat# MA1-2001; RRID:AB_ 326078 | IF (1:100) |
| Antibody | IgG, HRP-linked antibody (horse anti-mouse polyclonal) | Cell Signaling | Cat# 7076; RRID:AB_330924 | WB (1:2500) |
| Antibody | IgG, HRP-linked antibody (goat anti-mouse polyclonal) | Cell Signaling | Cat# 7074; RRID:AB_2099233 | WB (1:2500) |
| Antibody | IgG H&L (Alexa Fluor 594) preadsorbed (goat anti-mouse polyclonal) | Abcam | Cat# ab150120; RRID:AB_2631447 | IF (1:200) |
| Antibody | IgG H&L (Alexa Fluor 594) preadsorbed (goat anti-rabbit polyclonal) | Abcam | Cat# ab150084; RRID:AB_ 2734147 | IF (1:200) |
| Antibody | IgG H&L (Alexa Fluor 488) preadsorbed (goat anti-mouse polyclonal) | Abcam | Cat# ab150117; RRID:AB_ 2734147 | IF (1:200) |
| Antibody | IgG H&L (Alexa Fluor 488) preadsorbed (ab150081) (goat anti-rabbit polyclonal) | Abcam | Cat# ab150081; RRID:AB_2734747 | IF (1:200) |
| Antibody | IgG (H+L) Cross-Adsorbed Secondary Antibody, Alexa Fluor 488 (goat anti-mouse polyclonal) | Invitrogen | Cat# A-11001; RRID:AB_2534069 | IF (1:200) |
| Antibody | IgG (H+L) Cross-Adsorbed Secondary Antibody, Alexa Fluor 568 (goat anti-rabbit polyclonal) | Invitrogen | Cat# A-11011; RRID:AB_143157 | IF (1:200) |
| Recombinant DNA reagent | Dominant-negative (DN)-GSK3-GFP (*Xenopus laevis*) (plasmid) | Addgene | RRID:Addgene_29681 | K85R and K86R |
| Recombinant DNA reagent | DN-Rab7 (*Homo sapiens*) (plasmid) | Addgene | RRID:Addgene_12660 | Dominant-negative plasmid |
| Recombinant DNA reagent | pCS2-*mGFP* (plasmid) | Addgene | RRID:Addgene_14757 | Membrane bound form of EGFP |
| Recombinant DNA reagent | CD63-RFP (*Homo sapiens*) (plasmid) | Addgene | RRID:Addgene_62964 | RFP tag |
| Recombinant DNA reagent | xWnt8myc (wnt8a.L Frog) (plasmid) | Addgene | RRID:Addgene_16863 | |
| Recombinant DNA reagent | β-Catenin-activated reporter (BAR) (plasmid) | Addgene | RRID:Addgene_12456 | Beta-catenin reporter. TCF/LEF sites upstream of a luciferase reporter |

*Continued on next page*

Continued

| Reagent type (species) or resource | Designation | Source or reference | Identifiers | Additional information |
|---|---|---|---|---|
| Recombinant DNA reagent | Renilla reporter (plasmid) | Addgene | RRID:Addgene_62186 | MuLE (Multiple Lentiviral Expression) Entry vector containing a CMV promoter and renilla luciferase module |
| Recombinant DNA reagent | Lipofectamine 3000 | Thermo Fisher | Cat# L3000001 | |
| Peptide, recombinant protein | Wnt3a | Peprotech | Cat# 315-20 | |
| Commercial assay or kit | mMESSAGE mMACHINE SP6 Transcription Kit | Thermo Fisher | Cat# AM1340 | |
| Commercial assay or kit | Dual-Luciferase Reporter Assay System | Promega | Cat# E1500 | |
| Chemical compound, drug | Fibronectin | Thermo Fisher | Cat# 33016015 | |
| Chemical compound, drug | Dextran Tetramethylrhodamine (TMR-Dx) 70,000 | Thermo Fisher | Cat# D1818 | |
| Chemical compound, drug | 5-(N-Ethyl-N-isopropyl) amiloride (EIPA) | Sigma | Cat# A3085 | |
| Chemical compound, drug | Bafilomycin A1 | Selleckchem | Cat# S1413 | |
| Chemical compound, drug | Digitonin | Sigma | Cat# 300410 | |
| Chemical compound, drug | Phalloidin | Abcam | Cat# ab176759 | |
| Chemical compound, drug | Lithium chloride (LiCl) | Sigma | Cat# L4408 | |
| Chemical compound, drug | PF-00562271 (FAK inhibitor) | Selleckchem | Cat# S2672 | |
| Chemical compound, drug | Phorbol 12-myristate 13-acetate (PMA) | TOCRIS | Cat# 1201 | |
| Chemical compound, drug | 5-(N-Ethyl-N-isopropyl) amiloride (EIPA) | Sigma | Cat# A3085 | |
| Chemical compound, drug | EHT 1864 | Selleckchem | Catalog# S7482 | |
| Chemical compound, drug | Triton X-100 | Thermo Fisher | Cat# HFH10 | |
| Chemical compound, drug | Paraformaldehyde | Sigma | Cat# P6148 | |
| Chemical compound, drug | Fibronectin | Sigma | Cat# F4759 | |
| Chemical compound, drug | Protease inhibitors | Roche | Cat# 04693132001 | |
| Chemical compound, drug | Phosphatase inhibitors | Calbiochem | Cat# 524629 | |
| Software, algorithm | ImageJ | NIH | http://imagej.nih.gov/ij/ | |
| Software, algorithm | AxioVision 4.8 | Zeiss | http://Zeiss.com | |
| Software, algorithm | Zen 2.3 imaging software | Zeiss | http://Zeiss.com | |
| Software, algorithm | R | R Core Team | https://cran.r-project.org | |

*Continued*

| Reagent type (species) or resource | Designation | Source or reference | Identifiers | Additional information |
|---|---|---|---|---|
| Other | 10 cm dish | Thermo Fisher | Cat# 174903 | |
| Other | 8-Well glass-bottom chamber slides | ibidi | Cat# 80827 | |
| Other | Circular coverslips | ibidi | Cat# 10815 | |
| Other | 12-Well dish | Thermo Fisher | Cat# 150628 | |
| Other | DMEM (culture medium) | Thermo Fisher | Cat# 11965092 | |
| Other | L-15 (culture medium) | Thermo Fisher | Cat# 11415064 | |
| Other | Glutamine | Thermo Fisher | Cat# 25030081 | |
| Other | Fetal bovine serum (FBS) | Thermo Fisher | Cat# 16000044 | |
| Other | Bovine serum albumin (BSA) | Thermo Fisher | Cat# 9048468 | |
| Other | Pen-Strep antibiotics | Thermo Fisher | Cat# 15140122 | |
| Other | PBS | Gibco | Cat# 10-010-023 | |
| Other | PBS | Fisher Scientific | Cat# BP3994 | |
| Other | Fluoroshield Mounting Medium with DAPI | Abcam | Cat# ab104139 | |
| Other | IM 300 microinjection pump | Narishige International USA, Inc | N/A | |
| Other | Axio Observer Z1 Inverted Microscope with Apotome | Zeiss | N/A | |

## Materials availability statement

Further information and requests for resources and reagents should be directed to and will be fulfilled by the Lead Contact, Edward M. De Robertis.

No new materials were generated in this study. Requests for reagents used should be directed and will be fulfilled by the Lead Contact. No custom code, software, or algorithm central to supporting the main claims of the paper were generated in this manuscript.

## Experimental model and subject details

### Tissue culture and transfection

HeLa (ATCC, CRL-2648) and HEK293T cells stably expressing BAR and Renilla reporters were cultured in DMEM (Dulbecco's modified Eagle medium), supplemented with 10% fetal bovine serum (FBS), 1% glutamine, and penicillin/streptomycin. SW480 cells and SW480APC cells (*Faux et al., 2004*) were cultured in DMEM/F12 (DMEM:Nutrient Mixture F-12), supplemented with 5% FBS, 1% glutamine, and penicillin/streptomycin. The cells were seeded at a cell density of 20–30%, and experiments were performed when cells reached between 70 and 80% confluency. Cells were cultured for 6–8 hr in a medium containing 2% FBS before all treatments. All cell lines tested negative for mycoplasma contamination. ATCC cell line authentication services were performed to confirm identity of cultured cell types.

### *Xenopus* embryo microinjection

This study was performed in strict accordance with the recommendations in the Guide for the Care and Use of Laboratory Animals of the National Institutes of Health. All of the animals were handled according to approved Institutional Animal Care and Use Committee (IACUC) protocols (D16-00124) of the University of California, Los Angeles, Medical School. The protocol was approved by the Committee on the Ethics of Animal Experiments of the University of California, Los Angeles (Permit Number: ARC-1995-129). *Xenopus laevis* embryos were fertilized in vitro *using* excised testis and staged as described (*Colozza and De Robertis, 2014*; *Tejeda-Muñoz and De Robertis, 2022a*). In

vitro synthesized mRNAs were introduced into embryos by microinjection using an IM 300 Microinjector (Narishige International USA, Inc) of 4 nl into the marginal zone of a ventral blastomere at the 4-cell stage. pCS2-DN-GSK3β, β-catenin, and DN-Rab7 were linearized with NotI and transcribed with SP6 RNA polymerase using the Ambion mMessage mMachine kit. Embryos were injected in 1× MMR (Marc's Minimal Ringers) and cultured in 0.1× MMR.

## Human colon cancer tissue array and immunochemistry

Three sets of colon cancer tissue arrays containing 90 cases of adenocarcinoma and 90 adjacent normal colon tissue (180 tissue cores total) were obtained from TissueArray.com. The company stated that each specimen collected from any clinic was consented to by both the hospital and the individual, and that discrete legal consent forms were obtained and the rights to hold research uses for any purpose or further commercialized uses were waived. Double stained immunohistochemistry was performed on paraffin-embedded tissue samples which were deparaffinized in xylene and rehydrated using graded alcohols. For antigen retrieval, slides were incubated at 95°C for 40 min in citrate buffer (10 mM, 0.05% Tween 20, pH 6.0). Tissue sections were than fixed with 4% paraformaldehyde (Sigma #P6148) for 15 min, treated with 0.2% Triton X-100 in phosphate-buffered saline (PBS; Gibco) for 10 min, and blocked with 5% bovine serum albumin (BSA) in PBS overnight. Primary and secondary antibodies were added overnight at 4°C. The samples were washed three times with PBS after each treatment, and coverslips were mounted with Fluoroshield Mounting Medium with DAPI (ab104139). Immunofluorescence was analyzed and photographed using a Zeiss Imager Z.1 microscope with Apotome.

## *Xenograft* tumor model

For the CD1 NU/NU nude mouse model, SW480 cells were collected by trypsinization. Then, $7.5 \times 10^5$ cells per injection site were resuspended in high-concentration Matrigel (BD Biosciences, Franklin Lakes, NY), diluted in PBS to 50% final concentration, and subcutaneously injected into the flanks of an 8-week-old mouse. After day 7, when the tumor was established, the tumor was measured three times a week until completion. After 3 weeks, tissue was obtained from animals euthanized by $CO_2$ inhalation. Tumor and normal colon were fixed, paraffin-embedded, and 5 μm histological sections were mounted on slides. Mouse experiments were approved by the UCLA Animal Research Committee.

## Antibodies and reagents

Total β-catenin antibody (1:1000) was purchased from Invitrogen (712700), glyceraldehyde-3-phosphate dehydrogenase antibody (1:1000) and FAK antibody (1:1000, 3285) were obtained from Cell Signaling Technologies, anti-ATP6V0a3 antibody (1:500) was obtained from Novus (nbp1-89333, 1:1000). CD63 antibody was obtained from Abcam.

Antibodies against Pak1 (ab131522), Ras (ab52939), GSK3 (ab93926, 1:4000), and secondary antibodies for immunostaining for cells (ab150120, ab150084, ab150117, ab150081) (1:300) were obtained from Abcam. Antibody against the FA protein Tes (HPA018123, 1:100) was obtained from Atlas antibodies. Secondary antibodies for immunostaining arrays (A-11001, A-11011 were obtained from Invitrogen). HRP-linked secondary antibodies 7076, 7074 at 1:5000 (Cell Signaling) were used for western blots and analyzed with an iBright Imaging system. EIPA (A3085), and LiCl (L4408), were obtained from Sigma. Baf (S1413) and PF-00562271 (FAK inhibitor, S2672) were purchased from Selleckchem. TMR-dextran 70 kDa was obtained from Thermo Fisher (D1818). PMA (1201) was purchased from TOCRIS.

## **Immunostainings**

HeLa, HEK293T, SW480, and SW480APC cells were plated on glass coverslips and transferred to 2% FBS 6–12 hr before overnight experimental treatments. Coverslips were acid washed and treated with Fibronectin (10 μg/ml for 30 min at 37°C, Sigma F4759) to facilitate cell spreading and adhesion. Cells were fixed with 4% paraformaldehyde (Sigma #P6148) for 15 min, permeabilized with 0.2% Triton X-100 in PBS Gibco for 10 min, and blocked with 5% BSA in PBS for 1 hr.

Primary antibodies were added overnight at 4°C. Cells were washed three times with PBS, and secondary antibodies were applied for 1 hr at room temperature. After three additional washes with PBS, the coverslips were mounted with Fluoroshield Mounting Medium with DAPI (ab104139).

Immunofluorescence was analyzed and photographed using a Zeiss Imager Z.1 microscope with Apotome.

## Transfections

Transfections of β-cateninGFP (0.8 mg) and CA-Lrp6GFP (0.8 mg) were performed on 3T3 cells plated on coverslips for 24 hr. Cells were stained following the immunofluorescence protocol described above and stained with Rac-1 antibody.

## Western blots

Cell lysates were prepared using RIPA (Radioimmunoprecipitation Assay) buffer (0.1% NP40, 20 mM Tris/HCl pH 7.5), 10% glycerol, together with protease (Roche #04693132001) and phosphatase inhibitors (Calbiochem #524629), and processed as described (*Tejeda-Muñoz et al., 2019*).

## Luciferase assay

Experiments were performed with stably transfected HEK293T cells stably expressing BAR and Renilla reporters (*Albrecht et al., 2020*), treated with LiCl and with or without Bafilomycin for 8 hr, and Luciferase activity measured with the Dual-Luciferase Reporter Assay System (Promega) according to the manufacturer's instructions, using the Glomax Luminometer (Promega). Luciferase values of each sample were normalized for Renilla activity.

## 3D spheroid cell culture

SW480 cells were cultured in a Petri dish using DMEM:F-12 medium with 5% FBS. The top cover was removed from 60 mm tissue culture dishes and 3 ml of PBS placed in the bottom of the dish to act as a hydration chamber. Cells were counted and 500 cells were added as 25 µl drops deposited onto the Petri dish cover, and immediately inverted over the humid chamber. At least 6 drops per condition were plated, keeping enough distance between each other. The inverted drop cultures were incubated at 37°C in 5% $CO_2$/95% humidity.

The drops were monitored daily; after 4 days, aggregates had been formed and EHT treatment was added to the spheroids. After 4 days, spheroids were incubated with TMR-dextran 70 kDa (1 mg/ml) for 1 hr, and each spheroid was photographed with an Axio Zoom.V16 Stereo Zoom Zeiss microscope with apotome function.

## Time-lapse imaging

SW480 cells were seeded on glass bottom culture dishes (MatTek Corp). After 4 hr, the cells were transfected with mGFP with Lipofectamine 3000 (*Albrecht et al., 2020*). The video was taken after 3 days in culture. Control cells had DMSO added for 15 min. Then, cells were treated with PMA (0.3 µM) for 15 min. Filming was with a Zeiss Observer.Z1 microscope and AxioVision 4.8.2 SP3 software was used for imaging.

## TMR-dextran assays

3T3 or SW480 cells were seeded on cover glass coverslips for 12–18 hr as described in *Tejeda-Muñoz et al., 2022c*. The cells were incubated with Wnt3a overnight in the presence or absence of PMA and EIPA. After incubation, the cells were treated with 1 mg/ml TMR-dextran 70 kDa for 1 hr, then washed, fixed, and blocked for 1 hr with 5% BSA in PBS to reduce background from nonspecific binding. The coverslips were mounted using Fluoroshield Mounting Medium with DAPI (Abcam, ab104139). Immunofluorescence imagings were performed using a Zeiss Imager Z.1 microscope with Apotome.

## Bafilomycin and FAK inhibitor treatments, microinjection, and in situ hybridization in *Xenopus* embryos

We determined experimentally that optimal ventralization with loss of head structures of *Xenopus* embryos was obtained after incubation with Baf (5 µM) or FAK inhibitor PF-00562271 (100 µM) for 7 min at the 32-cell stage in 0.1× MMR solution. After incubation, embryos were washed two times with 0.1× MMR solution and cultured overnight until early tailbud tadpole stage. Embryos were microinjected once ventrally at the 4-cell stage with any of the following reagents: LiCl (300 mM), PMA (500 nM), EIPA (1 mM), EHT (1 mM), or mRNAs such as DN-GSK3-β (150 pg), DN-Rab-7 (500 pg), and

xWnt8 (2 pg), either alone or co-injected with the other reagents. In some experiments, microinjected embryos were incubated in 0.1× MMR solution until the 32-cell stage and transferred to 0.1× MMR solution with or without Baf (5 μM) and incubated for 7 min. Embryos were then washed twice with 0.1× MMR solution and cultured until tadpole stage. In situ hybridization was performed as described at http://www.hhmi.ucla.edu/derobertis.

### LiCl and Bafilomycin sequential treatments

Whole embryos at the 32-cell stage were incubated with LiCl (300 mM) alone for 7 min. After this treatment, the embryos were washed two times with 0.1× MMR solution and cultured until the tadpole stage, or incubated again for 7 min in 0.1 MMR solution containing 5 μM Baf, and washed as described before. In the converse experiment, embryos were first treated with Baf and then with LiCl. All treatments were done at the 32-cell stage, which plays a key role in the early Wnt signal (*Kao et al., 1986*; *Tejeda-Muñoz et al., 2022c*).

### qRT-PCR

Quantitative RT-PCR experiments using *Xenopus* embryos were performed as previously described (*Colozza and De Robertis, 2014*). Primer sequences for qRT-PCR were as follows:

| Gene | Forward | Reverse | Use |
|---|---|---|---|
| ODC | CAGCTAGCTGTGGTGTGG | CAACATGGAAACTCACACC | qRT-PCR |
| Siamois | AAGATAACTGGCATTCCTGAGC | GGTAGGGCTGTGTATTTGAAGG | qRT-PCR |
| Xnr3 | CGAGTGCAAGAAGGTGGACA | ATCTTCATGGGGACACGGA | qRT-PCR |
| Sizzled | GTCTTCCTGCTCCTCTGC | AACAGGGAGCACAGGAAG | qRT-PCR |
| Vent1 | GGCACCTGAACGGAAGAA | GATTTTGGAACCAGGTTTTGAC | |

### Quantification and statistical analysis

Data were expressed as means and standard error of the means (SEMs). Statistical analysis of the data was performed using the Student *t*-test; a p-value of <0.01** was considered statistically significant for differences between means. Fluorescence was quantified in control versus treated cells using ImageJ software analyses with $n > 30$ cells or $n > 30$ images from arrays from humans or mouse sample per condition. Fluorescence intensity was normalized in images compared in each condition and results from three or more independent experiments were presented as the mean ± SEM.

## Acknowledgements

We are grateful to Yi Ding and Yuki Moriyama for comments on the manuscript, and to Willy Wong for help depositing data. We are grateful for funding to the UC Cancer Research Coordinating Committee (grant C21CR2039); National Institutes of Health grant P20CA016042 to the University of California, Los Angeles Jonsson Comprehensive Cancer Center; startup funds from the University of Oklahoma; and the Norman Sprague Endowment for Molecular Oncology.

## Additional information

### Funding

| Funder | Grant reference number | Author |
|---|---|---|
| Cancer Research Coordinating Committee | C21CRC2039 | Edward M De Robertis |
| National Institutes of Health | P20CA016042 | Edward M De Robertis |

| Funder | Grant reference number | Author |
|---|---|---|
| Sprague Endowment for Molecular Oncology, University of California, Los Angeles | BD-55 | Edward M De Robertis |

The funders had no role in study design, data collection, and interpretation, or the decision to submit the work for publication.

## Author contributions

Nydia Tejeda-Munoz, Conceptualization, Data curation, Writing – original draft, Writing – review and editing; Yagmur Azbazdar, Data curation, Formal analysis, Investigation, Writing – review and editing; Julia Monka, Grace Binder, Alex Dayrit, Raul Ayala, Investigation; Neil O'Brien, Funding acquisition, Investigation; Edward M De Robertis, Conceptualization, Data curation, Funding acquisition, Investigation, Writing – original draft, Project administration, Writing – review and editing

## Author ORCIDs

Edward M De Robertis (iD) http://orcid.org/0000-0002-7843-1869

## Ethics

This study was performed in strict accordance with the recommendations in the Guide for the Care and Use of Laboratory Animals of the National Institutes of Health. All of the animals were handled according to approved Institutional Animal Care and Use Committee (IACUC) protocols (D16-00124) of the University of California, Los Angeles, Medical School. The protocol was approved by the Committee on the Ethics of Animal Experiments of the University of California, Los Angeles (Permit Number: ARC-1995-129).

Reviewer #1 (Public Review): https://doi.org/10.7554/eLife.89141.3.sa1
Reviewer #2 (Public Review): https://doi.org/10.7554/eLife.89141.3.sa2
Author Response https://doi.org/10.7554/eLife.89141.3.sa3

# Additional files

## Supplementary files

• MDAR checklist

## Data availability

Data have been deposited in the Cell Image Library (CIL): http://cellimagelibrary.org/groups/55765.

The following dataset was generated:

| Author(s) | Year | Dataset title | Dataset URL | Database and Identifier |
|---|---|---|---|---|
| Tejeda-Muñoz N, Azbazdar Y, Monka J, Binder G, Dayrit A, Ayala R, O'Brien N, De Robertis EM | 2023 | The PMA phorbol ester tumor promoter increases canonical Wnt signaling via macropinocytosis | http://cellimagelibrary.org/groups/55765 | Cell Image Library, 55765 |

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
