## [Editor Report · eLife assessment]

Altogether, the strength of this **important** study is that it provides **compelling** evidence in several biological models, including *Xenopus* embryos, that Wnt3a increases macropinocytosis and that PMA increases this cellular response. This novel link between Wnt, focal adhesions, lysosomes, and macropinocytosis will be very interesting for cell and tumor biologists. In future work, it will be important to identify the underlying mechanism, i.e., the molecular node whereby this interaction occurs.

---

## [Referee Report · Reviewer #1 (Public Review)]

In this study, the authors utilise different chemical inhibitors and celular markers to examine the roles of macropinocytosis in WNT signalling activation in development (*Xenopus*), cell culture (3T3 cells) and cancer (CRC sections). Furthermore, they investigate the effect of the inflammation inducer Phorbol-12-myristate-13-acetate (PMA) in WNT signalling activation through macropinocytosis. The authors show (1) that PMA induces macropinocytosis-dependent WNT signalling activation, and (2) that CRC development correlates with increased levels and co-localisation of macropinocytosis components and b-catenin.

I found the analyses and conclusions compelling. Additional epistatic analyses could be done in the future to further disentangle the roles of macropinocytosis during WNT signalling activation, especially upon oncogenic alterations (e.g. in APC). The studies on CRC samples open interesting questions for specialists in tumour progression.

---

## [Referee Report · Reviewer #2 (Public Review)]

Tejeda Muñoz et al. investigate the intersection of Wnt signaling, macropinocytosis, lysosomes, focal adhesions and membrane trafficking in embryogenesis and cancer. Following up on their previous papers, the authors present evidence that PMA enhances Wnt signaling and embryonic patterning through macropinocytosis. Strikingly, PMA and Wnt ligand act synergistically to trigger macropinocytosis in fibroblasts. Proteins that are associated with the endo-lysosomal pathway and Wnt signaling are co-increased in colorectal cancer samples, consistent with their pro-tumorigenic action. The function of macropinocytosis is not well understood in most physiological contexts, and its role in Wnt signaling is intriguing. The authors use a wide range of models - *Xenopus* embryos, cancer cells in culture and in xenografts and patient samples to investigate several endolysosomal processes that appear to act upstream or downstream of Wnt. This broad approach has the downside that results are often validated only in a subset of biological systems and that experiments tend to lack of mechanistic depth. The connections between PMA, Wnt signaling, Rac stabilization, FAK signaling and macropinocytosis remain unclear. Nevertheless, the results provide intriguing insights into a novel connection of the tumor promoting agent PMA and Wnt signaling in development and cancer.

The authors demonstrate striking, additive effects of Wnt3a and PMA in inducing macropinocytosis in 3T3 cells (Fig. 1 K-P). In the APC-mutant colorectal cancer line SW480, the authors show that PMA treatment increases macropinocytosis (Fig. S1). While these data provide additional confirmation that PMA can trigger macropinocytosis, they do not address the role of Wnt signaling directly. This could be done by restoring APC function in SW480 cells, or by ectopically activating Wnt signaling in a CRC cell line that lacks activating mutations in the Wnt pathway. These experiments would help to strengthen the cancer angle and validate the connection between Wnt signaling and PMA in macropinocytosis induction in additional cell lines.

The authors conclude that PMA enhances Wnt signaling based on experiments in *Xenopus* embryos where co-treatment with PMA and the Wnt activator LiCl increases Wnt target gene expression. This is an interesting observation, but large parts of the paper focus on mammalian cells / cancer cells. It would be important to demonstrate the ability of PMA to enhance Wnt signaling in these contexts as well.

---

## [Author Response]

The following is the authors’ response to the original reviews.

**Reviewer #1 (Public Review):**
In this ms, Tejeda-Muñoz and colleagues examine the roles of macropinocytosis in WNT signalling activation in development (*Xenopus*) and cancer (CRC sections, cell lines and xenograft experiments). Furthermore, they investigate the effect of the inflammation inducer Phorbol-12-myristate-13-acetate (PMA) in WNT signalling activation through macropinocytosis. They propose that macropinocytosis is a key driver of WNT signalling, including upon oncogenic activation, with relevance in cancer progression.I found the analyses and conclusions of the relevance of macropinocytosis in WNT signalling compelling, notably upon constitutive activation both during development and in CRC.

Thank you.

However, I think this manuscript only partially characterises the effects of PMA in WNT signalling, largely due to a lack of an epistatic characterisation of PMA roles in Wnt activation. For example: 1- The authors show that PMA cooperate with (1) GSK3 inhibition in *Xenopus* to promote WNT activation, and (2) (possibly) with APCmut in SW480 to induce b-cat and FAK accumulation. To sustain a specific functional interaction between WNT and PMA, the effects should be tested through additional epistatic experiments. For example, does PMA cooperate with Wnt8 in axis duplication analyses? Does PMA cooperate with any other WNT alteration in CRC or other cell lines? Importantly, does APC re-introduction in SW480 rescue the effect of PMA? Such analyses could be critical to determine specificity of the functional interactions between WNT and PMA. This question could be addressed by performing classical epistatic analyses in cell lines (CRC or HEK) focusing on WNT activity, and by including rescue experiments targeting the WNT pathway downstream of the effects e.g., dnTCF, APC re- introduction, etc.

We agree that there was need for additional direct evidence of functional interactions of between macropinocytosis, Wnt signaling, and PMA beyond the previously provided target gene assays in *Xenopus* (now shown in Figure 1I) and luciferase assays in cultured cells (Figure 1J) which used LiCl and inhibition by Bafilomycin. We therefore carried out a new experiment using 3T3 cells, now shown in Figure 1K-P. Wnt3a protein increased the uptake of TMR-dextran 70 kDa, and PMA enhanced this response. The macropinocytosis inhibitor EIPA blocked induction of macropinocytosis by Wnt3a and PMA. These results were quantitated in Figure 1Q. We think this new experiment strengthens the main conclusion that the tumor promoter PMA increases macropinocytosis. Thank you.

1. While the epistatic analyses of WNT and macropinocytosis are clear in frog, the causal link in CRC cells is contained to b-catenin accumulation. While is clear that macropinocytosis reduces spheroid growth in SW480, the lack of rescue experiments with e.g., constitutive active b-catenin or any other WNT perturbation or/and APC re-introduction, limit the conclusions of this experiment.

We now provide new experiments in 3T3 cells treated with LiCl, overexpression of constitutively-active β-catenin and constitutively-active Lrp6 (Figure 4, panels I through L’’); the new results indicate that Wnt signaling activation increases protein levels of the macropinocytosis activator Rac1.

Minor comments:3- Different compounds targeting membrane trafficking are used to rescue modes of WNT activation (Wnt8 vs LiCl) in *Xenopus*.

The main goal of our experiments was to test the requirement of membrane trafficking for tumor promoter activity through the Wnt pathway. We therefore used PMA, and a variety of inhibitors such as EIPA (Na+/H+ exchanger, Figure 1I and Figure 3D), Bafilomycin A (Figure 1H), DN-Rab7 (Figure 3G) and EHT1864 (a Rac1 inhibitor, Figure 4G). One could argue that using a wide variety of membrane trafficking inhibitors is a plus.

4- The abstract does not state the results in CRC/xenografts

We have added a sentence to the abstract.

5- Labels of Figure 2E might be swap

Thank you for detecting this error, we now label the last two columns in Figure 2E correctly.

6- Figure 4i,j, 6 and s4 rely on qualitative analyses instead of quantifications, which underscores their evaluation. On the other hand, the detailed quantifications in Figure S3A-D strongly support the images of Figure 5

The quantifications of the previous Figure 4I-J supported the data in the initial reviewed preprint, shown in Author response image 1:

However, these data have now been deleted from this version to make space for new experiments showing the stabilization of Rac1 by stabilized β-catenin and CA-LRP6. Quantifications in Figure 6C-F’’ are not shown because they represent changes in subcellular localization, but a western blot is provided in Figure 6B. Quantifications for Figure 6H-I’’ are shown in panel 6G. Supplemental Figure S4 already has 24 panels so introducing quantifications would be unwieldy.

Thank you for the thoughtful comments.

**Reviewer #2 (Public Review):**
Tejeda Muñoz et al. investigate the intersection of Wnt signaling, macropinocytosis, lysosomes, focal adhesions and membrane trafficking in embryogenesis and cancer. Following up on their previous papers, the authors present evidence that PMA enhances Wnt signaling and embryonic patterning through macropinocytosis. Proteins that are associated with the endo-lysosomal pathway and Wnt signaling are co-increased in colorectal cancer samples, consistent with their pro-tumorigenic action. The function of macropinocytosis is not well understood in most physiological contexts, and its role in Wnt signaling is intriguing. The authors use a wide range of models - *Xenopus* embryos, cancer cells in culture and in xenografts and patient samples to investigate several endolysosomal processes that appear to act upstream or downstream of Wnt. A downside of this broad approach is a lack of mechanistic depth. In particular, few experiments monitor macropinocytosis directly, and macropinocytosis manipulations have pleiotropic effects that are open alternative interpretations. Several experiments are confirmatory of previous findings; the manuscript could be improved by focusing on the novel relationship between PMA-induced macropinocytosis and better support these conclusions with additional experiments.

New additional experiments focusing on the role of PMA are now provided.

The authors use a range of inhibitors that suppress macropinosome formation (EIPA, Bafilomycin A1, Rac1 inhibition). However, these are not specific macropinocytosis inhibitors (EIPA blocks an Na+/H+ exchanger, which is highly toxic and perturbs cellular pH balance; Bafilomycin blocks the V-ATPase, which has essential functions in the Golgi, endosomes and lysosomes; Rac1 signals through multiple downstream pathways). A specific macropinocytosis inhibitor does not exist, and it is thus important to support key conclusions with dextran uptake experiments.

We used a wide range of inhibitors because the main idea is to show that membrane trafficking is important in Wnt and PMA activity. We would like to point out that the current experimental definition in the field of macropinocytosis, despite any caveats, is the ability to block dextran uptake with EIPA. Because inhibitors may not be entirely specific, we think using a broad approach to target membrane trafficking might be a plus. We now provide in Figure 1K-Q a new experiment showing that Wnt3a protein treatment increases dextran uptake and PMA stimulates this macropinocytosis in 3T3 cells. EIPA inhibited dextran macropinocytosis in the presence of Wnt and PMA (Figure 1N and 1Q). We also provide a time-lapse video of the rapid macropinocytic vesicles induction by PMA in SW480 CRC cells in which the plasma membrane is tagged (Supplemental Movie S1).

The title states that PMA increases Wnt signaling through macropinocytosis. However, the mechanistic relationship between PMA-induced macropinocytosis and Wnt signaling is not well supported. The authors refer to a classical paper that demonstrates macropinocytosis induction by PMA in macrophages (PMID: 2613767). Unlike most cell types, macrophages display growth factor-induced and constitutive macropinocytic pathways (PMID: 30967001). It would thus be important to demonstrate macropinocytosis induction by PMA experimentally in *Xenopus* embryos / cancer cells. Does treatment with EIPA / Bafilomycin / Rac1i decrease the dextran signal in embryos? In macrophages, the PKC inhibitor Calphostin C blocks macropinocytosis induction by PMA (PMID: 25688212). Does Calphostin C block macropinocytosis in embryos / cancer cells? Do the various combinations of Wnts / Wnt agonists and PMA have additive or synergistic effects on dextran uptake? If the authors want to conclude that PMA activates Wnt signaling, it would also be important to demonstrate the effect of PMA on Wnt target gene expression.

We now provide a new experiment showing macropinocytosis induction of PMA experimentally in cancer cells. CRC SW480 cells, despite having a mutant APC, are able to respond to PMA by further increasing TMR-dextran 70 kDa uptake over background within 1 hour (now shown in Figure S1):

Investigating PKC and Calphostin C is outside of goals of this paper. With respect to final the point on the effect of PMA on Wnt target gene expression, this was shown in the context of the *Xenopus* embryo in Figure 1I (Siamois and Xnr3 are direct targets of Wnt).

**Author response image 2. sa3fig2:** 

The experiments concerning macropinosome formation in *Xenopus* embryos are not very convincing. Macropinosomes are circular vesicles whose size in mammalian cells ranges from 0.2 - 10 µM (PMID: 18612320). The TMR-dextran signal in Fig. 1A does not obviously label structures that look like macropinosomes; rather the signal is diffusely localized throughout the dorsal compartment, which could be extracellular (or perhaps cytosolic). I have similar concerns for the cell culture experiments, where dextran uptake is only shown for SW480 spheroids in Fig. S2. It would be helpful to quantify size of the circular structures (is this consistent with macropinosomes?).

In response, we have deleted the TMR experiments in *Xenopus* embryos; they will be reinvestigated at a later time. With respect to macropinosome sizes in cultured cells, they are indeed large at the plasma membrane level (see new Supplemental Movie S1), but rapidly decrease in size once dextran is concentrated inside the cell. This can be visualized in the new experiments showing dextran vesicles in Supplemental Figure S1J-K and Figure 1K-P.

In Fig. 4I - J, the dramatic decrease in b-catenin and especially in Rac1 after overnight EIPA treatment is rather surprising. How do the authors explain these findings? Is there any evidence that macropinocytosis stabilizes Rac1? Could this be another effect of EIPA or general toxicity?

We now provide new evidence that Wnt signaling stabilizes Rac1. The old data relying on overnight EIPA treatment has been replaced by new experiments in 3T3 cells showing (i) that LiCl treatment increases levels of Rac1 protein and β-catenin levels (Figure 4I-J’’), (ii) that cells transfected with constitutively active β-catenin-GFP have higher levels of Rac1 than control untransfected cells (Figure 4K-K’’) and (iii) that Rac1 is stabilized in cells transfected with CA-Lrp6-GFP when compared to untransfected cells (Figure4L-L’’).

On a similar note, Fig. 6 K - L the FAK staining in control cells appears to localize to focal adhesions, but in PMA-treated cells is strongly localized throughout the cell. Do the authors have any thoughts on how PMA stabilizes FAK and where the kinase localizes under these conditions? Does PMA treatment increase FAK signaling activity?

The previous Figure 6K-L’’ are now found in Supplementary Figure S4, panels C-D’’. The result is that FAK is greatly stabilized by overnight incubation with PMA. How this achieved is unknown, perhaps the result of increased macropinocytosis, but we do not wish to speculate in the main manuscript. We have not measured FAK activity, but the FAK inhibitor PF-00562271 strongly decreased β-catenin signaling by GSK3 inhibition (Figure 6J) and has strong effects in neural development that mimic inhibition of the early Wnt signal (new experiments shown in Figure 6K-L’’’). The results suggest that FAK activity affects Wnt signaling and dorsal development; the molecular mechanism of this interaction is unknown but worthy of future studies.

The tumor stainings in Figure 5 are interesting but correlative. Pak1 functions in multiple cellular processes and Pak1 levels are not a direct marker for macropinocytosis. In the discussion, the authors discuss evidence that the V-ATPase translocates to the plasma membrane in cancer to drive extracellular acidification. To which extent does the Voa3 staining reflect lysosomal V-ATPase? Do the authors have controls for antibody specificity?

It is true that Pak1 has multiple functions, yet it is essential for the actin machinery that drives macropinocytosis. We have now rephrased the discussion to say “Rac1 is an upstream regulator of the Pak1 kinase required for the actin machinery that drive macropinocytosis (Redelman-Sidi et al., 2018)”. We also explain that: “V-ATPase has been associated with acidification of the extracellular milieu in tumors (Capecci and Forgac, 2013; Hinton et al., 2009; Perona and Serrano, 1988). Extracellular acidification is probably due to increased numbers of lysosomes which are exocytosed, since V0a3 was located within the cytoplasm in advanced cancer or xenografts in mice (Figures 5I and S3I)”. The antibody we used for V0a3 is highly specific and has been used widely (Ramirez et al., 2019).

**Reviewer #3 (Public Review):**
The manuscript by Tejeda-Munoz examines signaling by Wnt and macropinocytosis in *Xenopus* embryos and colon cancer cells. A major problem with the study is the extensive use of pleiotropic inhibitors as "specific" inhibitors of macropinocytosis in embryos. It is true that BafA and EIPA block macropinocytosis, but they do many other things as well. A major target of EIPA is the NheI Na+/proton transporter, which also regulates invasive structures (podosomes, invadopodia) which could have major roles in development. Similarly, Baf1 will disrupt lysosomes and the endocytic system, which secondary effects on mTOR signaling and growth factor receptor trafficking. The authors cannot assume that processes inhibited by these drugs demonstrate a role of macropinocytosis. While correlations in tumor samples between increased expression of PAK1 and V0a3 and decreased expression of GSK3 are consistent with a link between macropinocytosis and Wnt-driven malignancy, the cell and embryo-based experiments do not convincingly make this connection. Finally, the data on FAK and TES are not well integrated with the rest of the manuscript.

The criticism that drugs are not entirely specific is a valid one. Our approach of using a variety of drugs such as EIPA, BafA, EHT1864 or FAK inhibitor PF-00562271 all point to the main conclusion that the membrane trafficking is important in signaling by Wnt and the action of the tumor promoter PMA. The data on FAK, TES and focal adhesions have been better integrated in the manuscript and new experiments on the effect of FAK inhibitor in embryonic dorsal development are now provided (Figure 6K-L’’’).

1. The data in Fig. 1A do not convincingly demonstrate macropinocytosis - it is impossible to tell what is being labeled by the dextran.

In response, we have deleted the TMR-dextran experiments in *Xenopus* embryos; they will be reported at a later time.

1. The data in Fig. 2 do not make sense. LiCL2 bypasses the WNT activation pathway by inhibiting GSK3. If subsequent treatment with BafA blocks the effects of GSK3 inhibition, then BafrA is doing something unrelated to Wnt activation, whose target is the inhibition/sequestration of GSK3. While BafA might block GSK3 sequestration by inhibiting MVB function, it should have no effect on the inhibition of GSK3 by LiCl2.

We now explain in the main text describing Figure 2 in the results, the initial effect of GSK3 inhibition by LiCl is to trigger macropinocytosis (Albrecht et al., 2020). If the downstream acidification of lysosomes is inhibited, then the brief treatment with LiCl (7 min at 32-cell stage) has no effect (LiCl 1st+BafA 2nd, Figure 2H). BafA inhibits lysosomal acidification at 32-cell stage resulting in ventralization, but the effect of brief BafA treatment can be reversed by inducing membrane trafficking by LiCl (BafA 1st+LiCl 2nd, Figure 2C). The labelling of the figure panels C and H has been modified to indicate this is an order-of-addition experiment. These order-of-addition experiments strongly support the proposal that endogenous lysosomal activity is required to generate the initial endogenous Wnt signal that takes place at the 32-cell stage of development (Tejeda-Muñoz and De Robertis, 2022a).

1. The effect of EHT on MP in SW480 cells is not clearly related to what is happening in the embryos. The nearly total loss of staining for Rac and β-catenin after overnight EIPA does not implicate MP in protein stability - critical controls for cell viability and overall protein turnover are absent. Inhibition of WNT signaling might be expected to enhance β-catenin turnover, but the effect on Rac1 is surprising. A more quantitative analysis by western blotting is required.

The results from SW480 cells inhibition by EIPA have been replaced in Figure 4. We now provide new evidence in 3T3 cells that Wnt signaling stabilizes Rac1. The old data relying on EIPA treatment in SW480 cells has been replaced by new experiments in 3T3 cells showing (i) that LiCl treatment increases levels of Rac1 protein and β-catenin levels (Figure 4I-J’’), (ii) that cells transfected with constitutively active β-catenin-GFP have higher levels of Rac1 than control untransfected cells (Figure 4K-K’’) and (iii) that Rac1 is stabilized in cells transfected with CA-Lrp6-GFP when compared to untransfected cells (Figure4L-L’’). In the original EIPA experiment in SW480 cells, now deleted from this version of the manuscript, we tested the cell viability using a Vi-Cell Beckman-Coulter Viability Analyzer and found that cells were 96-98% viable but proliferation was strongly decreased after 12 h of EIPA treatment. The effect of brief Rac1 inhibition (7 min) in decreasing dorsal development in embryos at the critical 32-cell stage is robust (Figure 4A-C). In addition, coinjection of EHT is able to entirely block the effects of microinjected xWnt8 mRNA (compare Figure 4E to 4G, see also Figure 4H), suggesting that Rac1 is required for Wnt signaling. Quantitative target gene expression analysis is provided for the embryo experiments (Figure 4C and 4H); for the stabilization of Rac1 by Wnt we are not providing quantitative measurements, but found similar results with 3 independent approaches (LiCl, CA-β-catenin and CA-Lrp6).

1. The data on FAK inhibition and TES trafficking are poorly integrated with the rest of the paper.

We attempted to better relate the TES trafficking to our previous paper showing that canonical Wnt signaling induces focal adhesion and Integrin-β1 endocytosis. We now write in the results: “We have previously reported a crosstalk between the Wnt and focal adhesion (FA) signaling pathways. Wnt3a treatment rapidly led to the endocytosis of Integrin β1 and of multiple focal adhesion proteins into MVBs (Tejeda-Muñoz et al., 2022). FAs link the actin cytoskeleton with the extracellular matrix (Figure 6A), and we now investigated whether FA activity is affected by Wnt signaling, PMA treatment and CRC progression”.

**Reviewer #3 (Recommendations For The Authors):**
The reliance on pleiotropic inhibitors is a weakness and should be supplemented by genetic approaches to inhibit macropinocytosis.

We agree, but that would be outside of the scope of this study.